# The use of digital technology in non-pharmacological cognitive and psychosocial interventions for people with dementia and mild cognitive impairment: A scoping review

Ana B. Vivas[1], Imran Khan[2,3], Qarin Lood[4,5,6], Pierre Gander[2], Mia Dong[7], Simon Nielsen[7], Robert Lowe [2,8]*

**1** Neuroscience Research Center (NEUREC), CITY College, University of York Europe Campus, Thessaloniki, Greece, **2** Department of Applied IT, University of Gothenburg, Gothenburg, Sweden, **3** Department of Computer Science, University of Warwick, Coventry, United Kingdom, **4** Department of Health and Rehabilitation, Institute of Neuroscience and Physiology, Sahlgrenska Academy, Göteborg, Sweden, **5** University of Gothenburg Centre for Person-Centred Care (GPCC), University of Gothenburg, Gothenburg, Sweden, **6** School of Nursing and Midwifery, Faculty of Health Sciences, La Trobe University, Melbourne, Australia, **7** Brain+, Copenhagen, Denmark, **8** RISE AB, Gothenburg, Sweden

* robert.lowe@gu.se, robert.lowe@ri.se

## Abstract

### Objective

The purpose of this scoping review was to identify, geographically map, and understand what and how novel digital technologies are being used for providing cognitive training and psychosocial interventions that aim at preserving cognitive and everyday function, and improving well-being, in dementia patients, and persons with Mild Cognitive Impairment (MCI).

### Design

Studies were identified across seven databases (EMBASE, Scopus, CINAHL Complete, PubMed, Dimensions, APA PsychInfo) dating from 2012-Dec 2025, and one hundred and fourteen met the inclusion criteria. We followed the Joanna Briggs Institute guidelines and the framework. Screening was undertaken according to the PRISMA-SCR guidelines and involved 7 reviewers across stages.

### Results

This review analysed 114 articles and uncovered 8 main categories of digital technology focused on intervention delivery methods that adhered to our inclusion/exclusion criteria. Of those, gamified technology, apps, and web-based approaches were most commonly studied. An emerging technology identified was virtual reality based delivery. There was great heterogeneity in the evidence of effectiveness of the technology-based interventions on cognitive, functional and wellbeing outcomes

**Data availability statement:** All relevant data are within the paper and its Supporting Information files.

**Funding:** The authors received a budget from Forte (REF: 2021-01781) – "Digitalized Non-pharmacological Interventions for People with Dementia: Reviewing Scandinavian and International Contexts", (budget 996,300SEK) to carry out this scoping review research based on an expert panel-reviewed competitive grant application. The funders had no role in study design, data collection and analysis, decision to publish, or preparation of the manuscript.

**Competing interests:** NO authors have competing interests.

as reported in the studies included here. There was also great variability across the studies in how the interventions were designed and implemented and what and how many outcomes were measured.

## Conclusions

This review supports that the majority of the digital interventions for cognitive and psychosocial interventions are being developed and tested for administration under real-time supervision by a trained professional and not for unsupervised home environments. Notwithstanding, technology suitable for home use such as gamified tasks, apps and web/internet-based approaches were found to be the most prominent interfaces for digital interventions. Virtual reality (VR) based interventions, e.g., using simple headsets, must also be considered as intervention tools with high potential for home use mitigating issues of limited mobility through cognitive-physical training (combining physical – exergaming – and cognitive training).

## Introduction

The global trend of rapid use of digital technology is transforming many aspects of our societies and, together with demographic change into an ageing society, is pushing governments and relevant stakeholders into rethinking healthcare provision overall, and in particular as it concerns older adults with neurodegenerative disorders. It is estimated that over 55 million people worldwide are living with dementia (People with Dementia – PwD): a number that is projected to escalate to up to 78 million by 2030 [1]. With increasing numbers of PwD comes an increasing challenge to provide sufficient support across multiple forms of healthcare provision, and to help mitigate the financial and societal impact of dementia. This issue has been recently brought to the forefront by the COVID-19 pandemic: highlighting the need to provide tailored and provider-specific forms of digital support and care (both on-site and home-based) for PwD and older adults with cognitive impairments (see [2]). To respond to this challenge, various digital technologies have emerged in recent years in an attempt to replace or supplement more traditional methods of diagnosis, intervention and general health monitoring [3]. For example, digital gamified tasks via smartphones and tablets for diagnosis [4] and long-term monitoring [5]; digital forms of psychosocial therapy (such as for Cognitive Stimulation Therapy [6]; telemedicine (e.g., [2]); virtual and augmented reality in cognitive assessment and training [7]; human-robot interaction for memory training scenarios [8], and cloud-based tools for daily monitoring and supporting cognitive and social function, e.g., smart homes [9].

### Literature review

The increasing interest in digital healthcare is reflected in the number of systematic reviews of the use of digital technology for healthcare provision that have been published in recent years. For example, [10] reviewed evidence regarding the usage

of digital gaming interventions in Psychiatry. With regard to dementia, two recent reviews focused on caregivers. [11] conducted a scoping review of evidence relating to technologies directed at improving the mental health of caregivers of people with dementia; [12] conducted a systematic review of studies on education/knowledge of caregivers for using digital technology developed for dementia patients. A small number of reviews have also focused on PwD and older adults with or without cognitive impairment. [13] reviewed the evidence on the use of mobile applications for dementia screening, and [14] reviewed the evidence on the use of specific digital technologies for cognitive assessment, monitoring and training in PwD. Specifically, they reviewed the evidence on mobile (smartphone and tablet) applications, wearable technology and smart home systems. More recently [15] conducted a scoping review of digital healthcare technologies for PwD and MCI, including diagnosis, intervention and support for caregivers. The authors observed that, since 2015, there has been an increase in studies addressing digital healthcare technologies. The review also concluded that the characteristics of the end-user need to be taken into account when considering the suitability and potential effectiveness of the available technologies. Although there have been a few literature reviews in the last decade on digital healthcare for dementia and/ or older adults, there is a need for a more focused review on digital intervention for PwD and MCI that provides more in-depth information regarding their implementation (e.g., end-user characteristics, environment) and that aim at capturing any novel technology developed or implemented in non-pharmacological interventions given the fast pace of technology development.

Specifically, what is lacking from the above reviews is contextual information that could help to identify gaps in development, research and implementation, and could permit a deeper understanding of how best to use, and further develop, these technologies for supporting or delivering non-pharmacological interventions. One such key contextual factor is the geographical distribution of technology development/use, given the profound differences in digital literacy across countries [16,17]. For example, in Europe, there are striking differences between Northern countries, where over 90% of older adults use the internet weekly, and Southern and Eastern countries where less than 50% of older adults report weekly internet use [18]. There are also important differences across countries regarding strategic national goals, infrastructure, incentive structures, and investment in digital health development and implementation [19]. Within the Scandinavian context, for example, the Swedish government in 2018 proposed a 350 million SEK grant for municipalities to make use of digital and assistive technologies for the social care of people with dementia, to help mitigate the impact of dementia (https://www.government.se/contentassets/0550da61a00f4fcda2e600fdb0fbd79e/lena-hallengren-articles-2018-2022.pdf). Another key contextual factor is the site of deployment of interventions (e.g., nursing home, day-care facility or an end-user's private home), which has important implications for scalability access, and adherence to treatment [20,21]. We already know that compliance with the treatment's prescription is a major issue in healthcare, with evidence indicating that about 50% of people with chronic diseases fail to fully comply [22], with older people seeing a greater risk for non-compliance [23]. Additionally, cognitive training, within the context of digital technology-based interventions for persons with MCI or dementia, is not always a focus of reviews (e.g., [24]). We know that adherence to long-term computerised cognitive training (CCT) with older adults is influenced by characteristics such as digital literacy (previous experience with computers) and by the site, which is closely linked to the presence or not of supervision by a health professional or caregiver [25]. In general, it is crucial to consider specific technologies in relation to adherence since they have the potential to improve adherence to cognitive interventions by enhancing engagement and providing means for supervision, particularly in home environments [26,27].

Furthermore, given the fast growth of research on healthcare technology development and application [15], which has been further accelerated by the recent pandemic, there is a need for continuously updating and summarising the research evidence. One approach to updating the existing evidence, taking into consideration recent reviews, is to focus on newer or novel technologies (e.g., virtual reality and robotics) or novel applications or developments of "older" (which may be already commercialised) technologies. For example, computerised cognitive training (CCT) using a desktop computer or web technology has been around for decades and there are already commercialised solutions for use at home or in

healthcare settings. However, some of the more traditional CCT technologies are now incorporating gamified aspects to increase engagement and adherence or are being tested for their potentially enhanced effects when administered in combination with newer brain stimulation technologies or with other digital/non-digital components that include socialising or the collaboration of caregivers.

## Objectives

To address the gaps above, we conducted a scoping review to identify, geographically map, and understand what and how digital technologies are being used for providing or/and supporting non-pharmacological interventions for dementia and mild cognitive impairment (MCI) that focus on cognitive training, cognitive interventions or psychosocial interventions. Contributing to the body of evidence provided by relevant recent reviews, we focused on newer or novel technologies, or novel developments/applications of more mature digital technologies. The focus on novelty requires as a first step a scoping review before questions of effectiveness befitting of a systematic review can be investigated. We also summarise the evidence taking into consideration contextual factors other than user characteristics to evaluate, and qualitatively summarise, the available evidence for the potential effectiveness of the technologies in the context of the interventions provided. Given the lack of a cure for dementia, we focused on cognitive training and psychosocial interventions that aim at: 1) maintaining, for as long as possible, cognitive and everyday function and overall well-being in dementia patients; 2) preventing or delaying the onset of dementia by intervening in people with Mild Cognitive Impairment (MCI), which is considered a clinical stage between healthy ageing and dementia [15]. Specifically, we examined the literature on digital technologies for providing or supporting different forms of cognitive and psychosocial interventions to improve the cognitive, everyday function and/or well-being outcomes of people with dementia and MCI; considering geographical distribution, type of technologies, site of deployment and service user characteristics. The present scoping review addressed the following research questions:

RQ1: What forms of novel digital technologies are being used in cognitive and psychosocial interventions for persons with MCI and dementia?

RQ2: Are particular technologies being implemented/investigated in specific environments (e.g., day-care centre vs home) and subgroups of patients (e.g., dementia vs MCI)?

RQ3: What is the geographical mapping of the use of the technologies? What are the gaps in the development/research/implementation of technologies in Europe and beyond?

RQ4: What evidence is available for the potential effectiveness of the different technologies within the context of the interventions for improving cognitive, functional or well-being outcomes?

## Method and materials

### Search strategy

According to the three step search strategy recommendation of JBI, our approach is as follows: *step 1* consisted of initial exploratory search 'seeding' (piloting) using different databases and gathering key words from two iterations of revised search string seeds; *step 2* entailed: a word frequency search (n-gram search) using the seed string from step 1; analysis of the keywords and index terms led to a second search string; *step 3* entailed snowballing in order to find additional relevant literature.

For step 1, through an initial use of controlled terms (for medical subject headings) the seven expert reviewers produced an initial broad candidate string appending free-text terms for technology. Two iterations were carried out in workshops with physical attendance of reviewers in order to discuss the use of terms and then revise them according to the records found. It followed the PCC method, defined, in the context of our scoping review as:

P = Population, C (I) = Concept (Intervention), C (T) = Context (Technology). We used MeSH terms (medical subject headings) in relation to P and C(I) and broad-based digital and technology terms used to seed an initial pilot search using Scopus. The initial string was:

("dementia" OR "Alzheimer*" OR "MCI" OR "mild cognitive impairment") AND ("cognitive training" OR "cognitive intervention" OR "psychosocial intervention*" OR "cognitive stimulation") AND ("digi*" OR "computer*" OR "web*" OR "technolog*")

where AND boolean operators were used to separate respective P, C(I) and C(T) terms.

For step 2, we then conducted word frequency analysis (1–2 words) on titles, abstracts and key words, using the using the Tera word frequency algorithm (https://tera-tools.com/word-freq), retrieved from the seed search using Scopus to identify additional terminology for the PCC elements. We then ran a second iteration of the Scopus search using an updated seed string that incorporated additional technology-related terms identified through the n-gram frequency analysis. The n-gram frequency output can be found in the S4 Supplementary Material. This iterative approach is consistent with the JBI manual for evidence synthesis.

Based on the above approach, the final search string used was:

("dementia" OR "Alzheimer*" OR "MCI" OR "mild cognitive impairment") AND ("cognitive training" OR "cognitive intervention" OR "psychosocial intervention*" OR "cognitive stimulation") AND ("digi*" OR "computer*" OR "web*" OR "technolog*" OR "robot*" OR "*game" OR "gam*" OR "vr" OR "virtual reality" OR "computeri#ed cognitive" OR "cct").

We ran additional searches using known terms like exergame, exergaming and serious games, but this did not affect the terms found in the n-gram frequency search.

The most relevant to the field, and comprehensive, databases were deployed for searching the key terms relevant to our scoping review (see Fig 1. "Identification"): EMBASE, Scopus, CINAHL Complete, PubMed, Dimensions, APA PsychInfo. The initial search took place in February 2022 and was completed in December 2025. Since we wanted to be sure that our search string would capture all the new technologies, we refined our search strategy during this long period between the searches by consulting with other experts and researchers in the technology and the dementia/cognitive impairment fields. For step 3, we also utilised (initial search) the snowballing technique of a single iterative search of papers cited in those that were identified in the above-mentioned databases. We used the software *ResearchRabbit* (www.researchrabbit.ai) for this purpose. The full initial electronic search strings can be found in the table in the S1 Table. Inter-rater agreement per stage, as checked in Covidence (Covidence 2022), achieved 84% for Title and Abstract screening (8 reviewer pairs) and 74% for Full text screening (6 reviewer pairs). Furthermore, 6 reviewers were used for extraction and 3 reviewers for final consistency checks.

## Eligibility criteria

Title and Abstract Screening: Our inclusion criteria were as follows: i) published in peer-reviewed academic journals with extended papers; ii) published from 2012–2025 (for initial search we selected a 2012–2022 10 year timespan as an arbitrary cut-off date for extracting novel technology-based papers); iii) papers with new empirical data on interventions (qualitative or quantitative); iv) published in English, Swedish, Norwegian, Danish or Spanish (consistent with the mother tongue languages of the authors of this scoping review); v) participants had a diagnosis of MCI or dementia (all types of dementia) based on international standards; vi) dementia/MCI diagnosis was not a comorbidity of other neurological conditions (e.g., Parkinson's disease); vii) included a cognitive or psychosocial non-pharmacological intervention method; viii) included a digital/technology development/deployment.

Full-text Screening and Data extraction: In the second screening phase (See Fig 1, "Screening: Reports assessed for eligibility") the references included from the title-abstract screening were further reviewed at the full-text level and the following exclusion criteria were employed: i) exclusion of non-intervention study: unless feasibility or usability study that are supposed to lead into full interventions; ii) low quality: not sufficient consideration of ethical practices,

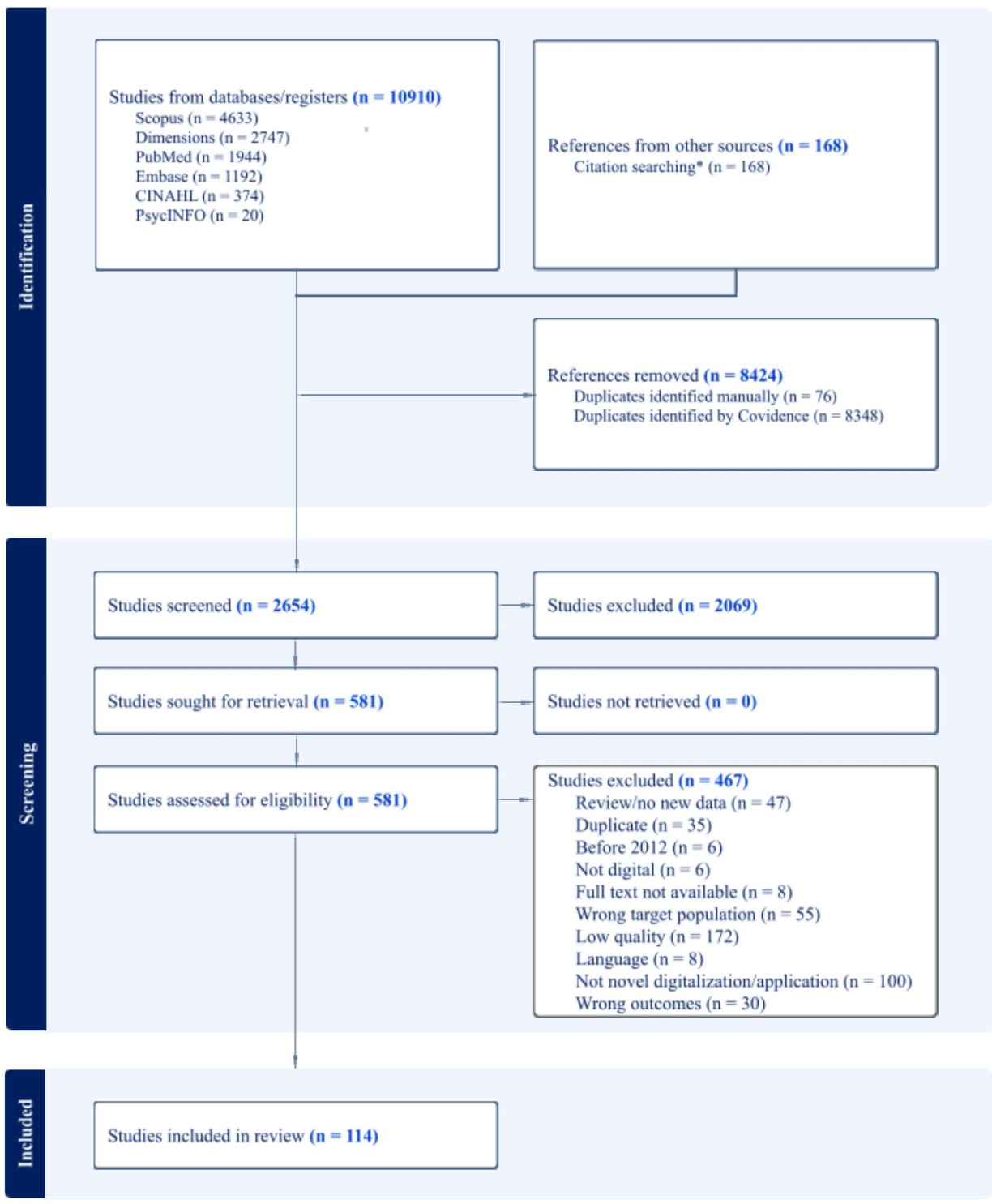

**Fig 1. Prisma flowchart of paper search and data extraction.**

criteria for diagnosis of dementia or MCI not specified or not in line with international guidelines, not full journal paper or extended conference paper; iii) wrong outcomes: outcomes that don't produce new data (as previously but with specific exclusion of prototypes, protocols) or outcomes that don't focus on either psychosocial, cognitive, well-being aspects; iv) wrong target population: participants did not have a diagnosis of MCI or dementia or dementia/MCI diagnosis was a comorbidity of other neurological conditions; v) not novel digital technology or development or application; the technology used to deliver/support the intervention is already commercialised and does not include any novel digital or application component to enhance the intervention, and vi) papers that are reviews or do not report new empirical data.

We adopted the below 3 criteria for evaluating whether the technology was to be excluded due to non-novelty. These criteria were decided based on a workshop carried out at the start of the project (February 2022):

1. Not novel digitalization or "novel application": e.g., beyond basic PC "interface" for Psychology task:

   • do not exclude "old" digitalization approach that has new application;

   • do not exclude "old" digitalization approach that is at increased maturation stage, e.g., new FDA approved example;

2. Not a temporary application: e.g., distance-based method, possibly due to the COVID-19 pandemic, and isn't intended as a longer term solution;

3. Not just a computer screen interface: digital technology can be software or hardware but not simply a computer screen unless novel software (or application) is used, i.e., point 1.

We define novel application as: technology that produces original data through a pilot study, intervention, validation study, or feasibility study, which aims to improve psychosocial function, cognition, or well-being in patients.

### Study selection

The references identified from the title and abstract screening search phase and after removing duplicates using Covidence, were n = 2654 (see Fig 1). Six of the authors (QL, SN, PG, AV, MD, IK) were involved in this review, and a pair of two reviewers reviewed each full-text reference. Another author (RL) resolved all the conflicts.

### Data extraction

We followed the checklist of the PRISMA-SCR (Preferred Reporting Items for Systematic reviews and Meta-Analyses extension for Scoping reviews) of [28] which entails adherence to 20 items when reporting, as characterised by Title, Abstract, Introduction, Methods, Results, Discussion and Funding. We were also guided by the JBI (Joanna Briggs Institute) method for conducting scoping reviews (e.g., [29]). The software Covidence (Covidence systematic review software, 2022), was employed for screening and extracting the data from the papers identified through the search.

Data extraction was guided by the JBI extraction template (https://jbi-global-wiki.refined.site/space/MANUAL/4687700) with additional data extracted in line with our research questions. All the authors (QL, SN, PG, AV, IK, MD, RL) were involved in the data extraction. The PRISMA diagram in Fig 1, depicts the search and extraction process carried out in this review.

Extracted papers (114) were listed by author(s) and categorised according to high-level categories of i) *Target population*; ii) *Number of participants*; iii) *Design*; iv) *Intervention type*, which concerns either the type of intervention the study evaluated or the potential type of intervention the intended study (feasibility, prototype development) was for; v) *Deployment*, which concerns whether the technology was designed to be used for individuals or with others; vii) *Study setting*; viii) *Technology type* (see table in S2 Table).

## Synthesis

We used [30] scoping review framework – enhanced by Peters et al. (2015) [29]–, which involves three steps: a) conducting a systematic literature search of multiple bibliographic databases, b) using a set of inclusion/exclusion criteria to identify and select relevant papers, and c) presenting a narrative synthesis of findings referred to as 'charting the evidence'. A quality critical appraisal of the articles was not conducted as it is not a requirement of scoping reviews; nevertheless, we excluded papers that were of low quality (according to our inclusion/exclusion criteria) in the screening phases and did not include articles that were shortened versions (or without new data) of another included paper by the same authors, where we checked for populations used and results obtained. We report conclusions and study findings following the checklist of the PRISMA-SCR (Preferred Reporting Items for Systematic reviews and Meta-Analyses extension for Scoping reviews) by [28].

We tabulate and summarise in succinct narratives the content of each intervention. We include a narrative description of the key patterns and observations and produce to summarise the study-level evidence so that we answer our research questions.

Finally, we followed the Clinical Studies Review **PICO Framework** to synthesize the evidence:

• **Patient, Population or Problem**

  • What are the characteristics of the patient or population (demographics, risk factors, pre-existing conditions, etc)?

  • What is the condition or disease of interest?

• **Intervention**

  • What is the intervention under consideration for this patient or population?

• **Comparison**

  • What is the alternative to the intervention (e.g., placebo, different drug, surgery)

• **Outcome**

  • What are the included outcomes (e.g., quality of life, change in clinical status, morbidity, adverse effects, complications)?

We adhered to PICO according to both search and eligibility criteria in the following way:

P(opulation): This was covered by our search string: ("dementia" OR "Alzheimer*" OR "MCI" OR "mild cognitive impairment")

I(nterventions): This was covered by the search string: ("cognitive training" OR "cognitive intervention" OR "psychosocial intervention*" OR "cognitive stimulation") AND ("digi* OR computer* OR web* OR technolog*")

C(omparator): This was covered by our eligibility criteria, where studies were only included that produced new data after an intervention. In these instances, comparator groups were pre/post-intervention groups, or another baseline group (e.g., healthy populations or another control group). Since our review was not explicitly interested in reviewing the effects of any specific intervention(s), we did not explicitly define comparator groups in our search strategy.

O(utcomes): This was covered by our screening process, where we were only interested in interventions that report on cognitive, well-being, functional outcomes, or are designed as a first step towards reporting those outcomes in the future (e.g., feasibility, pilot studies).

## Results

The results of extraction from 2012 to Dec 2025 indicate a general trend towards an increasing number of publications per year, peaking in 2024 (14 papers) – see Fig 2.

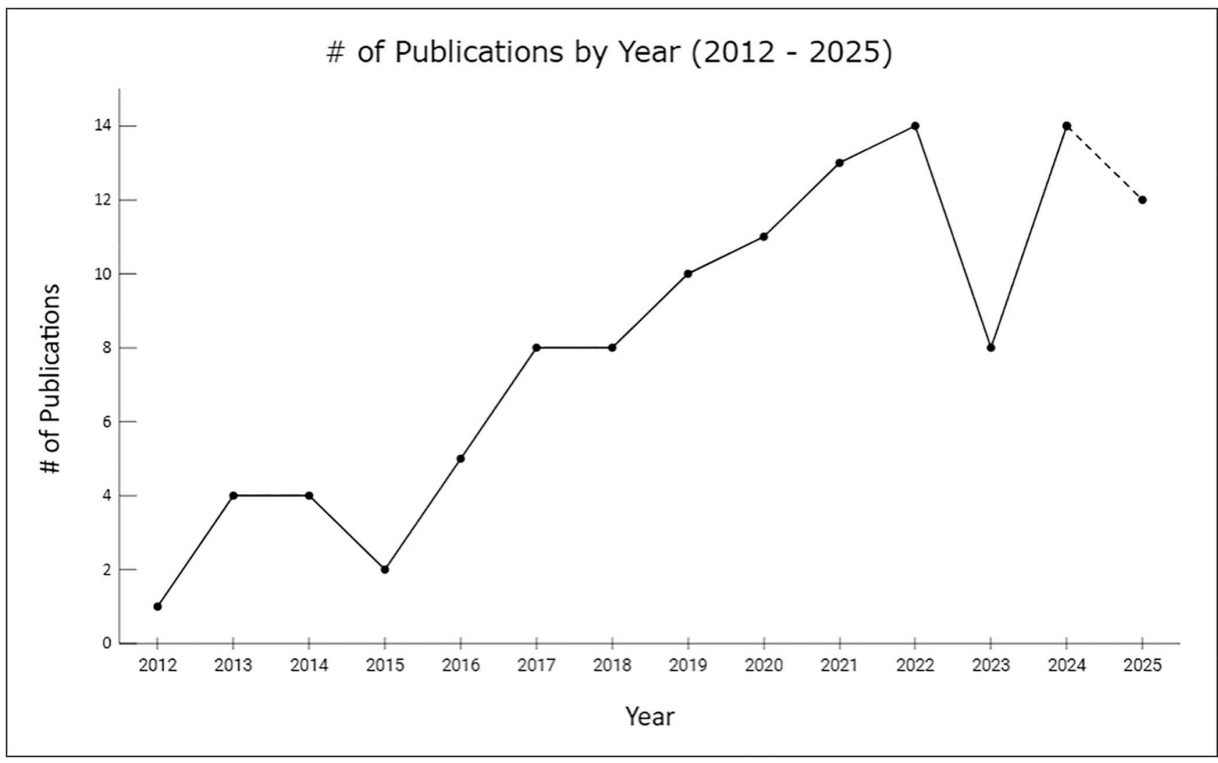

**Fig 2. The number of publications returned from the final screening, trended by year.**

It should be noted that as the initial literature search ("Identification" phase in Fig 1) was conducted on 7th Dec 2025, the number of publications reported for 2025 (12) does not include any papers that were published after the search date.

## Study characteristics

A summary of the study characteristics is presented in Table 1.

**Key (where not self-evident)**: *End Users/Population* – MCI (Mild Cognitive Impairment), aMCI (amnestic MCI), AD (Alzheimer's Disease) SMI (Subjective Memory Impairment), SCD (Subjective Cognitive Decline), VaDementia (Vascular Dementia), CI (Cognitive Impairment), ND (Neurological Disorder), ESD (Early Stage Dementia), HC (Healthy Control); *Study Design* – RCT (Randomized Control Trial), NRES (Non-Randomized Experimental Study), MPED (Mixed-Pairs Experimental Design); *Intervention Type* – CT (Cognitive Training), tDCS (transcranial Direct Current Stimulation), BCI (Brain-Computer Interface), rTMS (repetitive Transcranial Magnetic Stimulation); *Type of Technology* – DESK (Desktop), GT (Gamified Technology), ROB (Robotics);*Setting* – NH (Nursing Home), CLIN (Clinic), H (Home), DC (Daycare centre), NS (Not Specified), LAB (Laboratory).

Across all extracted studies, the majority were publicly funded (64.9%, n = 74). A smaller proportion were classified as non-sponsored (6.1%, n = 7) or industry-sponsored (5.3%, n = 6). Funding details were not reported in 11.4% of studies (n = 13) A considerable minority (12.3%, n = 14) received support from other non-public, non-industry sources, including registered charities, private foundations, university research funds, and hospital-affiliated grants. Two studies (1.7%) reported joint public–industry funding.

Table 2 presents the definitions of technology that we used to categorise the technologies.

**Table 1. Final extracted papers and characteristics.**

| Study | Ref | Country | End users/ Population | Total N (N intervention group/s) | Study design | Intervention type |
|---|---|---|---|---|---|---|
| Abdalrahim et al. (2022) | [31] | Jordan | Non-ESD | 60 (60) | Feasibility | Psychosocial; Reminiscence therapy |
| Alves et al. (2019) | [32] | Portugal | Dementia | 12 (12) | Design/ Development | Reminiscence therapy; Cognitive Stimulation |
| Amjad et al. (2019) | [33] | Pakistan | MCI | 44 (22) | RCT | CT; Physical |
| An et al. (2025) | [34] | South Korea | MCI | 88 (44) | RCT | CCT |
| Bahar-Fuchs et al. (2017) | [35] | Australia | MCI | 43 (21) | RCT | CT; Multidomain |
| Baik et al. (2023) | [36] | South Korea | MCI | 50 (25) | RCT | CCT |
| Baldimtsi et al. (2023) | [37] | Greece | MCI | 122 (28) | RCT | CT |
| Bamidis et al. (2015) | [38] | Greece | MCI; Dementia; HC | 322 (237) | NRES | CT |
| Barban et al. (2016) | [39] | Italy | MCI; AD; HC | 348 (348) | RCT | CT; Reminiscence therapy |
| Ben-Sadoun et al. (2016) | [40] | France | MCI; AD; HC | 18 (10) | Feasibility | CT |
| Bernini et al. (2023) | [41] | Italy | mNCD; SCD | 14 (10) | Feasibility | CT |
| Brem et al. (2020) | [42] | United States, Italy | AD-mild-moderate | 49 (16) | RCT | CT; rTMS |
| Burgos-Morelos et al. (2025) | [43] | Mexico | MCI; Dementia-mild | 38 (38) | Quasi-experimental study | CCT |
| Caroppo et al. (2017) | [44] | Spain | Dementia-mild-moderate | 12 (12) | Feasibility | CST |
| Cavallo et al. (2016) | [45] | Italy | AD | 80 (40) | RCT | CT |
| Chae et al. (2024) | [46] | South Korea | Dementia | 60 (30) | RCT | CT; Multidomain |
| Chandler et al. (2019) | [47] | USA | MCI | 272 (272) | RCT | CT; Multidomain |
| Chantanachai et al. (2024) | [48] | Australia | Dementia-mild-moderate | 61 (31) | RCT | CT |
| Choi et al. (2025) | [49] | South Korea | MCI | 17 (17) | Feasibility | CCT |
| Christogianni et al. (2022) | [50] | UK; India | MCI | 31 (16) | NRES | CCT |
| Çinar et al. (2020) | [51] | Turkey | AD; SCI | 120 (30) | NRES | CT; Physical; Pharmachological; Lifestyle |
| Combourieu Donnezan et al. (2018) | [52] | France | MCI | 75 (40) | RCT | CT; Physical/Exergaming |
| Cruz et al. (2013) | [53] | Portugal | Dementia | 80 (20) | Feasibility | CT |
| Danesin et al. (2025) | [54] | Italy | MCI | 32 (16) | RCT | CCT |
| De Luca et al. (2016) | [55] | Italy | ESD | 20 (10) | NRES | CT |
| Dethlefs et al. (2017) | [56] | United Kingdom | Dementia | 23 (10) | Feasibility | CST |
| Diaz Baquero et al. (2022) | [57] | Spain | ESD; Non-ESD | 89 (57) | RCT | CT |
| Djabelkhir-Jemmi et al. (2018) | [58] | France | aMCI; Non-aMCI | 51 (51) | NRES | CT; CST |

*(Continued)*

**Table 1.** (Continued)

| Study | Ref | Country | End users/ Population | Total N (N intervention group/s) | Study design | Intervention type |
|---|---|---|---|---|---|---|
| Duff et al., (2022) | [59] | United States | aMCI | 146 (72) | RCT | CT |
| Ferreira et al. (2023) | [60] | Portugal | Dementia-all stages | 8 (8) | Pilot study | CST |
| Fiatarone Singh et al. (2014) | [61] | Australia | MCI | 100 (51) | RCT | CT; Physical |
| Finn et al. (2014) | [62] | Australia | aMCI | 2 (2) | Case Study | CT |
| Fiorini et al. (2019) | [63] | Italy | MCI; HC | 53 (53) | Feasibility | CT; Physical |
| Gaitán et al. (2013) | [64] | Spain | MCI; ESD | 60 (37) | RCT | CT |
| Gandelman-Marton et al. (2017) | [65] | Israel | Dementia-mild | 8 (8) | Pilot; Feasibility | CT; rTMS |
| Gigler et al. (2013) | [66] | United States | aMCI; HC | 18 (18) | Feasibility; NRES | CT |
| Givon Schaham et al. (2024) | [67] | Israel | MCI | 61 (30) | RCT | CT |
| Gonzalez et al. (2021) | [68] | Hong Kong | Non-aMCI | 67 (22) | RCT | CT; tDCS |
| Gonzalez-Palau et al. (2014) | [69] | Spain | Non-aMCI | 50 (50) | Feasibility; NRES | CT |
| Graessel et al. (2024) | [70] | Germany | MCI | 89 (44) | RCT | CCT |
| Hagovská et al. (2017) | [71] | Slovakia | Non-aMCI | 60 (30) | RCT | CT |
| Han et al. (2024) | [72] | South Korea | MCI | 20 (20) | NRES; Feasibility | CCT |
| Han et al. (2017) | [73] | South Korea | Non-aMCI | 50 (25) | RCT | CT |
| Han et al. (2020) | [74] | South Korea | Non-aMCI | 24 (24) | RCT; Feasibility | CST |
| Hang et al. (2019) | [75] | China | AD | Not specified | Feasibility | CT |
| Harvey et al. (2024) | [76] | United States | aMCI; Non-aMCI, multi-domain MCI | 92 (47) | RCT | CT |
| Harvey et al. (2020) | [77] | United States | Non-aMCI | 95 (50) | RCT | CT |
| Hassandra et al. (2021) | [78] | Greece | Non-aMCI | 57 (57) | RCT; Feasibility | CT |
| Hird et al. (2024) | [79] | Japan | Dementia | 15 (15) | Pilot | Cognitive Stimulation |
| Hoel et al. (2022) | [80] | Germany | Dementia | 18 (18) | NRES | Social Interaction |
| Hung et al. (2020) | [81] | China | ESD | 10 (10) | Feasibility | CT |
| Hwang et al. (2023) | [82] | Taiwan | MCI | 189 (51) | RCT | CT |
| Hyer et al. (2016) | [83] | United States | Non-aMCI | 68 (34) | RCT | CT |
| Jeong et al. (2025) | [84] | South Korea | AD-mild-moderate | 13 (13) | NRES | CT |
| Kaanan et al. (2014) | [85] | United States | AD | 21 (21) | Cohort study | CT |
| Kang et al. (2024) | [86] | South Korea | MCI | 29 (15) | RCT | CCT |
| Karssemeijer et al. (2019) | [87] | Netherlands | Dementia-mild-moderate | 115 (38) | RCT | CT; Physical |
| Kim et al. (2019) | [88] | South Korea | MCI | 44 (31) | NRES | CT |
| Kim et al. (2021) | [89] | South Korea | aMCI; Non-aMCI | 15 (15) | Cohort study | CT |
| Latella et al. (2024) | [90] | Italy | MCI | 50 (50) | Feasibility; quasi-experiment | CT |
| Lau et al. (2024) | [91] | Taiwan | MCI | 22 (11) | RCT | CT; tDCS |
| Lee et al. (2023) | [92] | South Korea | MCI | 15 (15) | NRES | CT |

*(Continued)*

**Table 1.** (Continued)

| Study | Ref | Country | End users/ Population | Total N (N intervention group/s) | Study design | Intervention type |
|-------|-----|---------|---------|---------|---------|---------|
| Lee et al. (2020) | [93] | South Korea | MCI | 46 (24) | Pilot RCT | CCT |
| Lee et al. (2018) | [94] | South Korea | ESD | 75 (52) | RCT | CT |
| Li et al. (2024) | [95] | China | MCI; Dementia-mild-moderate | 60 (30) | NRES | CT |
| Liang et al. (2017) | [96] | New Zealand | Dementia | 60 (30) | RCT | Social Interaction |
| Liao et al. (2019) | [97] | Taiwan | MCI | 42 (21) | RCT | CT; Physical (exergaming) |
| Lin et al. (2021) | [98] | United States | aMCI | 49 (24) | RCT | CCT |
| Lin et al. (2022) | [99] | Taiwan | aMCI; Non-aMCI | 24 (12) | NRES | CCT |
| Lissek et al. (2024) | [100] | Germany | MCI | 47 (47) | NRES | CT; Multidomain |
| Maeng et al. (2021) | [101] | South Korea | MCI | 56 (31) | NRES | CT |
| Manca et al. (2021) | [102] | Italy | MCI | 14 (14) | NRES | CT |
| Manenti et al. (2020) | [103] | Italy | MCI | 49 (49) | RCT | CT; Cognitive Stimulation |
| Manenti et al. (2024) | [104] | Italy | MCI | 109 (109) | RCT | CT; Cognitive Stimulation |
| Manser et al. (2024) | [105] | Switzerland | MCI | 41 (21) | RCT | CCT |
| Marin et al. (2022) | [106] | USA | MCI; Dementia-mild | 19 (10) | Feasibility; RCT | CT |
| Mendoza Laiz et al. (2018) | [107] | Spain | aMCI; Non-aMCI | 32 (32) | Case Study | CT; BCI |
| Mondellini et al. (2022) | [108] | Estonia | MCI | 15 (15) | Feasibility | CT |
| Montero-Odasso et al. (2023) | [109] | Canada | MCI | 175 (141) | RCT | CT; Physical; Nutrition-based |
| Na et al. (2025) | [110] | South Korea | MCI | 32 (32) | Pilot Feasibility | CCT; Physical |
| Na et al. (2018) | [111] | South Korea | aMCI; Non-aMCI; SMI | 26 (16) | Case Study | CT |
| Navarro et al. (2018) | [112] | Mexico | Dementia | 7 (7) | Feasibility | CST |
| Nieto-Vieites et al. (2024) | [113] | Spain | MCI; SCD | 22 (11) | Feasibility; MPED | CT |
| O´Sullivan et al. (2022) | [114] | Germany | AD; VaDementia; unspecified Dementia; other CI | 162 (80) | RCT | Cognitive stimulation; emotion regulation |
| Oh et al. (2018) | [115] | South Korea | AD; VaDementia | 30 (15) | RCT | CT; BCI |
| Oliveira et al. (2021) | [116] | Portugal | AD | 18 (10) | RCT | CT; Cognitive stimulation |
| Park (2022a) | [117] | South Korea | aMCI | 50 (25) | RCT | CT |
| Park (2022b) | [118] | South Korea | MCI | 56 (56) | RCT | CCT |
| Park et al. (2019) | [119] | South Korea | MCI | 21 (21) | RCT | CT |
| Pino et al. (2012) | [120] | France | MCI | 22 (11) | Usability | CT; social support |
| Prinz et al. (2025) | [121] | Germany | Dementia | 61 (61) | Pilot RCT | CCT |
| Rai et al. (2021) | [122] | UK | Dementia-mild-moderate | 122 (62) | RCT; Feasibility | CST |
| Robert et al. (2020) | [123] | France | AD; mixed ND | 46 (25) | RCT | CT |
| Rouse et al. (2019) | [124] | United States | Dementia | 10 (10) | RCT | CT; non-digital social interaction |
| Savulich et al. (2017) | [125] | UK | aMCI | 42 (21) | RCT | CT |
| Senczyszyn et al. (2023) | [126] | Poland | MCI | 38 (26) | RCT | CT; rTMS |
| Senczyszyn et al. (2021) | [127] | Poland | MCI; SCI | 91 (66) | NRCT | CT; Psychoeducation; social interaction; Whole-Body Stimulation |
| Shandera-Ochsner et al. (2021) | [128] | United States | aMCI | 544 (544) | RCT | CT; Multidomain |
| Shin et al. (2020) | [129] | South Korea | MCI; healthy controls | 80 (40) | RCT | CT |

*(Continued)*

| Study | Ref | Country | End users/ Population | Total N (N intervention group/s) | Study design | Intervention type |
|---|---|---|---|---|---|---|
| Shyu et al. (2022) | [130] | Taiwan | Dementia-mild-moderate | 30 (15) | RCT | CT |
| Tan et al. (2025) | [131] | Netherlands | Dementia-mild-severe | 81 (40) | RCT | CST; Psychosocial |
| Tsiakari et al. (2025) | [132] | Greece | MCI | 100 (50) | RCT | CCT |
| van Santen et al. (2020) | [133] | Netherlands | Dementia-mild-moderate | 112 (73) | RCT | CT; Physical |
| Wiloth et al. (2018) | [134] | Germany | Dementia | 99 (56) | RCT | CT; Physical |
| Wu et al. (2020) | [135] | Taiwan | Dementia-mild-moderate | 16 (not specified) | RCT | Psychosocial; Reminiscence therapy |
| Yang et al. (2022) | [136] | South Korea | MCI | 99 (33) | RCT | CCT |
| Yu et al. (2015) | [137] | Hong Kong | Dementia-mild-moderate | 32 (16) | RCT | CT |
| Yu et al. (2025) | [138] | China | MCI | 92 (66) | (two stage) RCT | CT; physical |
| Yun et al. (2020) | [139] | South Korea | Dementia-mild; MCI | 11 (11) | Feasibility | CT |
| Zaccarelli et al. (2013) | [140] | Greece, Italy, Norway, Spain | aMCI; ESD; AD; HC | 348 (348) | RCT | CT; Social Interaction |
| Zając-Lamparska et al. (2019) | [141] | Poland | Dementia-mild | 150 (75) | NRES | CT |
| Zhang et al. (2025) | [142] | China | MCI; AD-mild | 84 (42) | RCT | CCT |
| Zhu et al. (2022) | [143] | China | aMCI; Dementia-mild-moderate | 35 (35) | Feasibility | CT |
| Zuschnegg et al. (2025) | [144] | Austria | AD-mild-moderate | 22 (11) | RCT | CCT; Physical |

| Study | Ref | Dose (minutes x days x weeks) | Dropout Rate (%) | Adherence (%) | Technology Type(s) | Setting(s) |
|---|---|---|---|---|---|---|
| Abdalrahim et al. (2022) | [31] | NS | 25 | 80 | APP | NH |
| Alves et al. (2019) | [32] | NS | NS | NS | APP, WEB | NH, CLIN |
| Amjad et al. (2019) | [33] | 25-30m x 5d x 6w | 13.6 | NS | GT | HOSP |
| An et al. (2025) | [34] | 30 m x 5d x 10w | 0 | 100% | APP; GT | H |
| Bahar-Fuchs et al. (2017) | [35] | 20-30m x 3d x 8-12w | 24 | 81-100 | WEB | H |
| Baik et al. (2023) | [36] | ~24 m x 3d x 8w | 0 | NS | APP | H |
| Baldimtsi et al. (2023) | [37] | 20-45mx2–3dx12w | 0 | NS | VR, APP | DC |
| Bamidis et al. (2015) | [38] | NS | 29 | NS | WEB | H,DC, NH |
| Barban et al. (2016) | [39] | 60m x 2d x 12w | 8 | NS | DESK | DC |
| Ben-Sadoun et al. (2016) | [40] | ~ 45m x 3d x 4w | 0 | NS | DESK; GT | NS |
| Bernini et al. (2023) | [41] | 45m x 3d x 6w | 0% | 100% | DESK | H |
| Brem et al. (2020) | [42] | 60m x 5d x 6w | 4 | NS | APP, BCI | LAB, CLIN |
| Burgos-Morelos et al. (2025) | [43] | 20-30m x 3d x 12w | NS | NS | DESK; VR | HOSP |
| Caroppo et al. (2017) | [44] | 15-20m x 1d x 1w | 0% | NS | GT; APP; WEB | LAB |
| Cavallo et al. (2016) | [45] | 30 m x 3d x 12w | 5 | 100 | DESK | NH |
| Chae et al. (2024) | [46] | 30-50mx5dx8w | 0 | NS | APP, GT | H |
| Chandler et al. (2019) | [47] | 240mx5dx2w | 16.2 | NS | APP | H |

*(Continued)*

| Study | Ref | Dose (minutes x days x weeks) | Dropout Rate (%) | Adherence (%) | Technology Type(s) | Setting(s) |
|---|---|---|---|---|---|---|
| Chantanachai et al. (2024) | [48] | 20-90m x 1-3d x 24weeks | 12.9 | 90 | WEB, GT | H |
| Choi et al. (2025) | [49] | --m x 5-7d x 4w | 0 | 122.35% | APP; GT | H |
| Christogianni et al. (2022) | [50] | --m x 7d x 4w | 0 | NS | APP; GT | H |
| Çinar et al. (2020) | [51] | 15-20m x 7d x 12w | 14.9 | NS | APP, WEB | H |
| Combourieu Donne-zan et al. (2018) | [52] | 60m x 2d x 12w | 12.5 | 81.2 | GT | NS |
| Cruz et al. (2013) | [53] | 20m x 1d x 1w | 40 | NS | GT; WEB | HOSP |
| Danesin et al. (2025) | [54] | 45 m x 5d x 2w | 0 | 100 | WEB | HOSP |
| De Luca et al. (2016) | [55] | 45m x 3d x 8w | 0 | NS | WEB | NH |
| Dethlefs et al. (2017) | [56] | 20m x 1d x 1w | 0% | NS | DESK | NS |
| Diaz Baquero et al. (2022) | [57] | 30m x 2-3d x 16w | 10 | NS | DESK | DC, NH, HOSP |
| Djabelkhir-Jemmi et al. (2018) | [58] | 90m x 2d x 12w | 12 | NS | WEB | CLIN |
| Duff et al. (2022) | [59] | 45m x 4-5d x 12-13w | 23.6 | NS | WEB | H |
| Ferreira et al. (2023) | [60] | 30-45m x 2d x 7w | 0 | 100% | VR | NH; DC |
| Fiatarone Singh et al. (2014) | [61] | 60-100m x 2d x 26w | 8 | 100 | DESK | LAB |
| Finn et al. (2014) | [62] | 90m x 2d x 6w | 0 | 100 | WEB | CLIN |
| Fiorini et al. (2019) | [63] | NS | 7.5 | NS | Foot sensor, APP | LAB |
| Gaitán et al. (2013) | [64] | 60m x 2-3d x 12w | 22.56 | NS | WEB | LAB |
| Gandelman-Marton et al. (2017) | [65] | 45m x 5d x 6w | 12.5 | NS | DESK | NS |
| Gigler et al. (2013) | [66] | 30m x 2d x 8-10w | 11.11 | NS | GT | H |
| Givon Schaham et al. (2024) | [68] | 30-60mx3–5dx5w | 16.6 | 94.4 | APP | H |
| Gonzalez et al. (2021) | [68] | 30m x 3d x 3w | 2.99 | 100 | DESK | LAB |
| Gonzalez-Palau et al. (2014) | [69] | 35m x 3d x 12w | 12 | NS | WEB | DC |
| Graessel et al. (2024) | [70] | ~30 m x 3d x 24w | 9 | NS | DESK | H |
| Hagovská et al. (2017) | [71] | 30m x 2d x 10w | 3.33 | NS | GT | CLIN |
| Han et al. (2024) | [72] | 20-30m x 5d x 12w | 5 | 100% | APP; GT | H |
| Han et al. (2017) | [73] | 30m x 2d x 4w | 18 | NS | APP | H |
| Han et al. (2020) | [74] | 60m x 2d x 10w | 29.41 | NS | DESK | LAB, CLIN |
| Hang et al. (2019) | [75] | NS | 0 | NS | VR | LAB |
| Harvey et al. (2024) | [76] | 2d x 12w | 3.6 | 92 | DESK, WEB | DC |
| Harvey et al. (2020) | [77] | 30m x 2d x 12w | 9 | NS | DESK | LAB |

*(Continued)*

| Study | Ref | Dose (minutes x days x weeks) | Dropout Rate (%) | Adherence (%) | Technology Type(s) | Setting(s) |
|---|---|---|---|---|---|---|
| Hassandra et al. (2021) | [78] | one session | 0 | NS | VR | LAB |
| Hird et al. (2024) | [79] | 30m x 1d x 1w | 0% | NS | APP | NH |
| Hoel et al. (2022) | [80] | 15m x 3d x 4w | 22 | 100 | APP | H |
| Hung et al. (2020) | [81] | --m x 2d x 5w | 20 | NS | APP | LAB |
| Hwang et al. (2023) | [82] | 50m* x 1d x 24w | NS | 75 | DESK, WEB | H |
| Hyer et al. (2016) | [83] | 40m x 5d x 5-7w | 13.24 | NS | DESK | LAB, H |
| Jeong et al. (2025) | [84] | 30m x 2d x 9w | 8 | 93% | VR | H |
| Kanaan et al. (2014) | [85] | 240-300m x 8–9 d x 2w | 0 | 63 | DESK | DC |
| Kang et al. (2024) | [86] | 30m x 2d x 8w | 0 | NS | APP | CLIN |
| Karssemeijer et al. (2019) | [87] | 30-50m x 3d x 12w | 11 | 87.30% | VR | LAB |
| Kim et al. (2019) | [88] | --m x 2-3d x 6w | 0 | NS | DESK | NS |
| Kim et al. (2021) | [89] | 50-60m x 2d x 4w | 29 | NS | VR | HOSP |
| Latella et al., (2024) | [90] | 45m x 2d x 20w | 0 | NS | VR | CLIN |
| Lau et al. (2024) | [91] | 40m x 3d x 5 w | 0 | 100 | VR, GT | LAB |
| Lee et al. (2023) | [92] | -- x 1dx4w | 13.33 | NS | VR | LAB |
| Lee et al. (2020) | [93] | 60m x 5d x 4w | 16.7 | 96.90% | ROB | H |
| Lee et al. (2018) | [94] | 30m x 2d x 6w | 0 | NS | DESK | NS |
| Li et al. (2024) | [95] | 60m x 1d x 12w | 0 | NS | VR | NH;HOSP |
| Liang et al. (2017) | [96] | 30m x 2-3d x 6w | 13 | 87 | ROB | DC, H |
| Liao et al. (2019) | [97] | 60m x 3d x 12w | 14.2 | NS | VR | DC |
| Lin et al. (2021) | [98] | 300m x --d x 4w | 14 | NS | APP | NH |
| Lin et al. (2022) | [99] | 60m x 1d x 12w | 33 | 100 | GT | DC |
| Lissek et al. (2024) | [100] | 60mx2dx12w | 36.17 | NS | APP, GT | LAB |
| Maeng et al. (2021) | [101] | 50-60m x 2d x 4w | 23 | NS | VR | NS |
| Manca et al. (2021) | [102] | --m x --d x 4w | 0 | 100 | ROB | CLIN |
| Manenti et al. (2020) | [103] | 60m x 3d x12w | NS | NS | VR | CLIN, H |
| Manenti et al. (2024) | [104] | 60m x 3d x 12w | 35 | NS | VR; BCI | CLIN, H |
| Manser et al. (2024) | [105] | 24m x 5d x 12w | 10 | 90% | GT | CLIN, H |
| Marin et al. (2022) | [106] | 30 m x 7d x 24w | 20 | 80 | APP | H |
| Mendoza Laiz et al. (2018) | [107] | 120m x 2d x 5w | 0 | 100 | WEB | DC |
| Mondellini et al. (2022) | [108] | 20m* x 1d x 1w | 0% | NS | VR | CLIN |
| Montero-Odasso et al. (2023) | [109] | 90m x 3d x 20w | 18 | 87 | APP | LAB |
| Na et al. (2025) | [110] | 50m x 2d x 12w | 12.5 | NS | VR | HOSP |
| Na et al. (2018) | [111] | 40 m x 2d x12w | 0 | 100 | DESK | CLIN |
| Navarro et al. (2018) | [112] | 45m x 1-2d x 12w | NS | NS | GT, WEB | DC |
| Nieto-Vieites et al. (2024) | [113] | 60m x 12d | 0% | NS | GT | LAB |
| O´Sullivan et al. (2022) | [114] | 30m x 3d x 8w | 30 | 15 | APP | NH |

*(Continued)*

**Table 1.** (Continued)

| Study | Ref | Dose (minutes x days x weeks) | Dropout Rate (%) | Adherence (%) | Technology Type(s) | Setting(s) |
|---|---|---|---|---|---|---|
| Oh et al. (2018) | [115] | 40m x 3d x 6w | 0 | NS | GT | NH |
| Oliveira et al. (2021) | [116] | 45m x 2d x 6w | 5 | NS | VR | NH |
| Park (2022a) | [117] | 45m x 3d x 8w | 0 | NS | VR | DC |
| Park (2022b) | [118] | 30 m x 3 d x 8 w | 0 | 100 | VR | LAB |
| Park et al. (2019) | [119] | 30m x 1d x 6w | NS | NS | VR | LAB |
| Pino et al. (2012) | [120] | NS | NS | NS | ROB | NS |
| Prinz et al. (2025) | [121] | 10m x 3d x 10w | 42.6 | NS | GT | NH |
| Rai et al. (2021) | [122] | 30m x 2-3d x 11w | 8.5 | 11 | APP | H |
| Robert et al. (2020) | [123] | 30m x 4d x 12w | 0 | 41 | WEB, APP | H |
| Rouse et al. (2019) | [124] | --m x 1-3d x7w | 20 | NS | GT, WEB | DC |
| Savulich et al. (2017) | [125] | 60m x 2d x 4w | 0 | 100 | GT, APP | LAB |
| Senczyszyn et al. (2023) | [127] | 3mx5dx2w | 15.3 | NS | WEB | LAB |
| Senczyszyn et al. (2021) | [126] | 90m x 1d x 9w | 7.6 | NS | WEB | LAB |
| Shandera-Ochsner et al. (2021) | [128] | 45-60m x~5d x 2w | 0 | NS | GT, APP | LAB, DC |
| Shin et al. (2020) | [129] | 30m x 3d x 4w | 10 | 100 | APP, GT | HOSP |
| Shyu et al. (2022) | [130] | 30m x 1d x 12w | 33 | NS | WEB, GT | H |
| Tan et al. (2025) | [131] | 30m x 2d x 4w | 25 | NS | WEB | NH |
| Tsiakari et al. (2025) | [132] | 45m x 1d x 24w | NS | NS | GT | CLIN |
| van Santen et al. (2020) | [133] | 30m x 2-5d x 24w | 29 | NS | GT | DC |
| Wiloth et al. (2018) | [134] | 90m x 2d x 10w | 16 | 86.3 | GT | LAB |
| Wu et al. (2020) | [135] | 90m x 1d x 2w | 0 | NS | WEB | CLIN |
| Yang et al. (2022) | [136] | 100 m x 3d x 8w | 0 | 94% | VR | LAB |
| Yu et al. (2015) | [137] | 30m x 1-2d x 4-8w | 6.25 | NS | GT, APP | LAB |
| Yu et al. (2025) | [138] | 60m x 2d x 24w | 18 | NS | VR; APP | DC |
| Yun et al. (2020) | [139] | 30m x 1d x 1w | 0% | NS | VR | NS |
| Zaccarelli et al. (2013) | [140] | 60m x 2d x 12w | NS | NS | GT, APP | LAB |
| Zając-Lamparska et al. (2019) | [141] | 45-60mx2dx4w | 66.6 | NS | VR, GT | NS |
| Zhang et al. (2025) | [142] | 45 m x 5d x 24w | 11.9 | NS | WEB | H |
| Zhu et al. (2022) | [143] | 20-30m x 3d x 5w | 11.1 | 100 | VR, GT | DC |
| Zuschnegg et al. (2025) | [144] | ~23m x~4d x 24w | 8 | NS | APP | H |

It should be noted that FOOT is used in a single study [63], and so the description for this technology relates specifically to the device used in that study. CT = Cognitive Training; CS = Cognitive Stimulation.

With regard to the populations targeted by the technologies in these studies (see Table 3), most studies involved people with mild cognitive impairment (PwMCI). Across the 114 included studies, 66.7% 5% (n = 76) involved PwMCI, while 43.9% (n = 50) involved people with dementia (PwD). This included 12 (10.54%) studies that implemented the intervention with both groups (e.g., [38,39,43,106,139,140,143,145]).

**Table 2. Types of technologies used in the extracted articles with descriptions given for categorization purposes.**

**Definition of Digital Technologies**

| Type | Abbreviation | Description |
|---|---|---|
| Game or Gamified Task | GT | A game or task designed to provide cognitive training or stimulation that includes game-like elements or has a game-based focus |
| Mobile or Tablet Application | APP | Technology that allows delivery of intervention via smart devices (e.g., mobile phone or tablet) |
| Laptop or Desktop-Based Software | DESK | Tasks or exercises to provide CT or CS that do not include game aspects and are administered via a computer. |
| Web/Browser-Based Application | WEB | Applications that are accessible on a browser from home or on-site (day-care centre, clinic, etc.) and typically entail an app not specific to mobile or tablet, and enable providing an intervention remotely. They often integrate different aspects (e.g., CT, exergaming, psychosocial, data monitoring). |
| Virtual Reality | VR | Virtual Reality (VR) equipment with or without head-mounted display (HMD). This includes immersive and 'non-immersive' VR, Augmented Reality and Mixed Reality. |
| Robot | ROB | A physical (rather than simulated) robotic agent that is an important part of the intervention (e.g., social interaction). |
| Foot Sensor | FOOT | An inertial sensor that is worn on the participant's foot to track movement. |

**Table 3. Types of technologies used across all studies by population in all studies. Thirty-two studies included more than one technology type, and twelve studies included both patient groups.**

| End User Classification | Technology Type | | | | | | | | |
|---|---|---|---|---|---|---|---|---|---|
| | GT | APP | VR | WEB | DESK | ROB | BCI | FOOT | |
| MCI | 21 | 19 | 22 | 14 | 14 | 3 | 1 | 1 | 95 |
| Dementia | 15 | 17 | 10 | 15 | 9 | 1 | 1 | 0 | 68 |

Certain technologies were used more frequently with particular populations. Virtual reality (VR) interventions were more often tested with PwMCI (22 instances) than with PwD (10 instances), with five studies including both groups. A similar trend was observed for gamified tasks (GT), where more instances targeted PwMCI (21) than PwD (15). Overall, these trends show more focus of novel digital interventions with PwMCI, with comparatively fewer technologies specifically studied with dementia populations.

With regard to geographical distribution, the 114 studies covered within this scoping review are spread across 29 countries (see Table 4): 14 countries in Europe (Austria, Estonia, France, Germany, Greece, Italy, Netherlands, Norway, Poland, Portugal, Slovakia, Spain, Switzerland, and the UK), 10 in Asia (China, Hong Kong, India, Israel, Japan, Jordan, Pakistan, South Korea, Taiwan, Turkey), 3 in North America (Canada, Mexico, USA), and 2 in Oceania (Australia, New Zealand). No studies fitting the review criteria were found from Africa, or from Central or South America. Overall, 44 of the 114 studies (38.6%) were (partially or wholly) conducted in Asia: 23 studies in South Korea (the single country contributing the most studies), 6 in Taiwan, 6 in China, 2 in Hong Kong and 2 in Israel, and 1 each from Japan, Pakistan, India, Jordan, and Turkey. The United States of America and Italy each accounted for the second-highest number of studies at the country level (12 apiece). 51 studies were conducted in at least one European country. Only 10 of these were conducted in what might be considered non-Western European countries (Greece, 5; Poland, 3; Slovakia, 1; Estonia, 1). One of the five Greek studies [140] was a multi-site intervention that also involved Italy, Norway, and Spain. The European distribution of studies was dominated by Western European countries such as Italy (12 studies, including multi-country trials), Spain (7), Germany (6), France (5), and the UK (4). Of the Scandinavian countries, only Norway had a single study attributed to it, via the multi-site intervention reported by [140].

**Table 4. Geographical distribution of digital technologies. Number of technologies employed per country in the 114 studies included in this review.**

| Continent | Country | Number of technologies used over all studies |
|---|---|---|
| Asia | China | 7 |
| | Hong Kong | 3 |
| | Japan | 1 |
| | Pakistan | 1 |
| | South Korea | 26 |
| | Taiwan | 9 |
| | Israel | 2 |
| | Jordan | 1 |
| | Turkey | 2 |
| Europe | Estonia | 1 |
| | France | 7 |
| | Germany | 7 |
| | Greece | 7 |
| | Italy | 15 |
| | Norway | 2 |
| | Poland | 4 |
| | Portugal | 6 |
| | Slovakia | 1 |
| | Spain | 10 |
| | UK | 5 |
| North America | Canada | 1 |
| | Mexico | 3 |
| | USA | 16 |
| Oceania | Australia | 5 |
| | New Zealand | 1 |

Gamification or games-based interventions (GT) emerged as the most-investigated technology type globally, being used at least once in 21 of the 29 countries included in this review. GT was deployed in 6 of 10 Asian countries (all apart from Israel, Japan, Jordan, and Turkey) and in all but two of the European countries (Austria and Estonia). Mobile or tablet applications (APP) had the second-widest geographical spread, being implemented in 19 of 29 countries, with web-based technologies (WEB) and computer-based applications (DESK) following, being studied in 13 and 12 countries respectively. Ten different countries implemented virtual reality (VR). Studies using this technology were mostly concentrated in Asia, which accounted for 15 of the 27 VR instances: South Korea contributed 10 of these VR studies, with China (3) and Taiwan (2) accounting for the remainder in Asia. In Europe, six countries accounted for 10 VR applications: Italy (3), Greece (2), Netherlands (1), Estonia (1), Poland (1), and Portugal (2). Mexico (1) was the only country outside of Asia or Europe to apply VR in the included studies. Despite being classed as an emerging technology, the use of robots (ROB) was markedly limited in comparison, with only four studies implementing this technology in interventions for PwMCI or PwD: one each in New Zealand [96], France [120], Italy [102], and South Korea [93].

## Characteristics of digital interventions

From the main summary Table (see Table 1), we identified eight different types of technology used to provide cognitive and psychosocial interventions in PwD and MCI. The distribution of these technologies can be seen in Fig 3. A total of 148 technologies were used across the 114 studies extracted. Three main types of technologies accounted for the majority of the overall

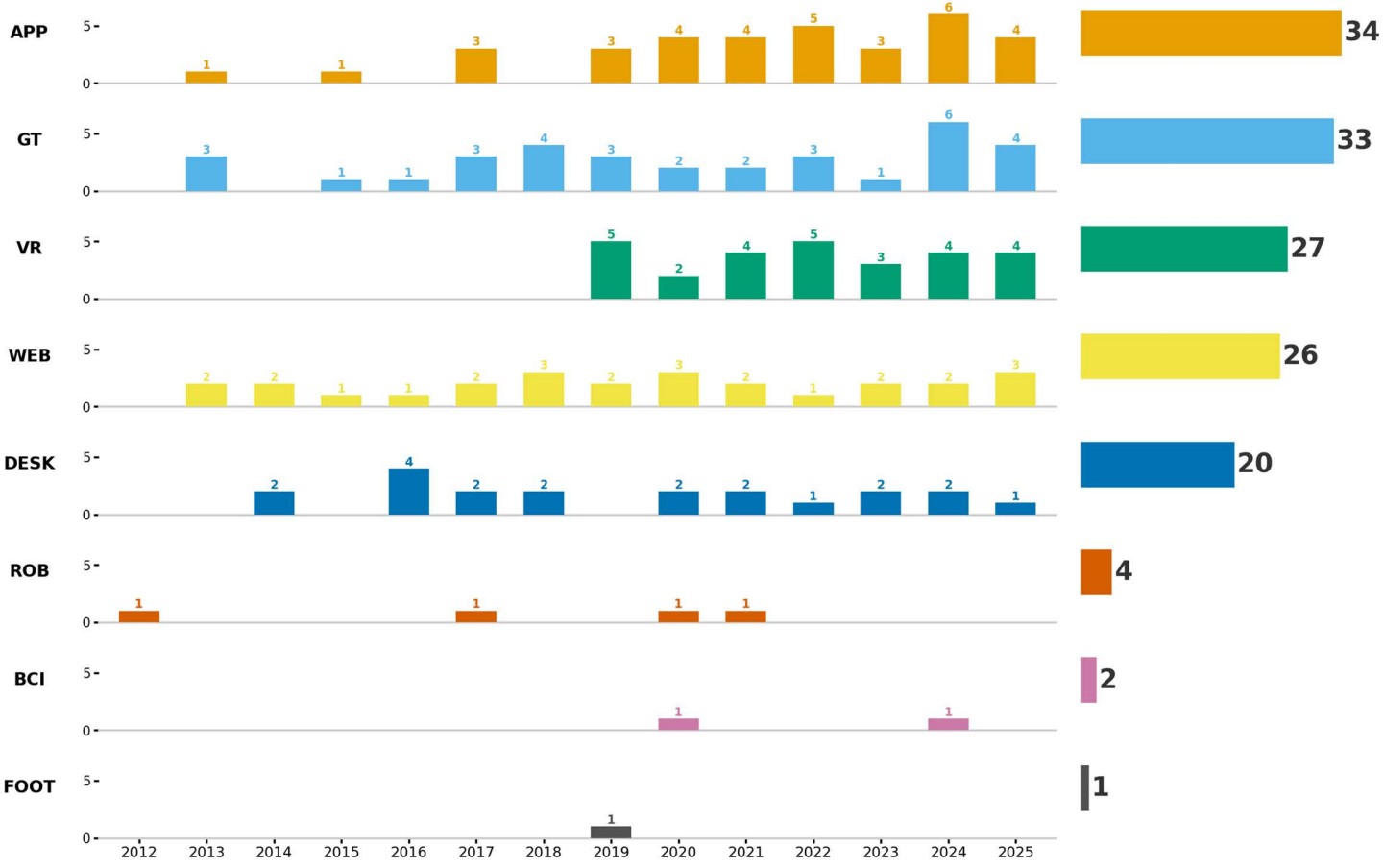

**Fig 3. Types of technology used over all settings.** GT = Game or gamified task. APP = Mobile or tablet-based application. WEB = Web-based digital technology. DESK = Desktop-based, laptop, or other computer-based application. VR = Virtual Reality. BCI = Brain-Computer Interface. ROB = Robot. FOOT = Foot Sensor..

technological usage (95/149, or 63.7%) in these studies: i) Mobile or tablet-based applications (23.1%), ii) Games or gamified tasks (22.51%), iii) virtual reality (18.4%). Web-based applications and desktop/laptop-based software accounted for 17.7% and 13.6% of all technology usage, respectively. However, it is important to point out that 32 of the 114 (28%) studies employed more than one technology, often combining technologies together. For example, [137] delivered a gamified task (GT) using a mobile touch-screen application (APP), [39] combined GT and APP in a multi-domain cognitive training programme across several European sites [43] integrated both VR and DESK-based components within a hybrid cognitive-motor training protocol.

An example of an intervention using Web-based (WEB) digital technology is the one described in [45], which used the "KODRO" platform. The KODRO platform is a web-based service that includes a range of cognitive exercises that target a range of cognitive domains. These cognitive exercises are designed to be ecologically relevant (i.e., the cognitive exercises relate to everyday activities in some way). The intervention in this study was set up as a group session (with persons with MCI) managed by a neuropsychologist, who was able to remotely set the individual difficulty levels for each participant, as well as monitor performance metrics and feedback offered by the platform, for each participant across all training sessions. In [38] a web-service system, which integrated and registered performance on physical and cognitive training (adaptation of *Posit Science* cognitive training), was deployed in an RCT with PwD and MCI. Physical training was administered using exergames utilising the *Wii Balance Board* or *Wii Remote* and involved aerobic ("hiking" and "cycling"

training) and anaerobic (strength-based exercises) and a number of balance-based training tasks (e.g., "Ski Jump"). A more recent example of a web-based intervention can be found in [54], who evaluated a structured multidomain cognitive training programme delivered through an online platform to older adults with MCI. Other web-based systems emphasised structured remote accessibility and automated session management. For example, [107] delivered a multidomain cognitive training programme via an online platform that allowed participants with MCI to complete scheduled sessions remotely, while performance metrics were automatically recorded.

Observably (see Fig 3), one of the technologies that continues to be consistently used is mobile or tablet-specific applications (APP). Concerning interventions using such technology, [106] used the Constant Therapy program, developed for patients with speech and/or cognitive deficits, for a home-based intervention with PwD. The Constant Therapy program was accessed through a touchscreen tablet device and consisted of a suite of cognitive exercises designed to improve a range of cognitive skills, which participants self-administered at their own pace throughout the week. These exercises increased in difficulty once participants achieved consistently high scores (3 x 80%+) on a particular exercise. This application also monitored exercise progress and time spent performing the cognitive tasks, which was remotely monitored by research staff who performed weekly check-ins with the participants. More recently, [49] developed a novel mobile gamified cognitive application incorporating adaptive difficulty scaling, reporting improvements in working memory and processing speed in PwMCI. Several of the studies extracted here also used tablet-based applications across a range of settings and intervention durations, including a 4-week intervention in senior independence living centres [98] in PwMCI, a 12-week intervention in nursing home residents with dementia [31], an 8-week multidomain gamified cognitive training programme delivered at home [50], and a 6-month tablet-based multimodal intervention for Alzheimer's disease in [144]. The latter combined cognitive and physical mini-games delivered entirely at home, with weekly supervised check-ins. Likewise, [86] used a mobile/tablet application to deliver multidomain CCT to people with MCI in a memory-clinic setting, while [93] integrated five tablet-based cognitive-training programs into a home-deployed social robot platform, demonstrating the flexibility of application delivery across different combinations of technologies as well as context. Other APP interventions emphasised feasibility and real-world usability. For example, [86] examined tablet-based cognitive training in residential dementia care environments, focusing on adherence and staff-supported implementation, and [109] integrated cognitive exercises with motor dual-task components to address gait-cognition interactions, as an approach towards functional, multidomain digital rehabilitation.

Several examples of a novel interventions using a gamified task/games (GT) can be found in [44,53] and [143]. In the latter study, the authors created and tested a gamified task using a virtual supermarket environment, deployed using a virtual reality (VR) headset. Here, participants were asked -to memorise a short list of everyday items, then to engage in a number sorting task for 20 seconds, before "entering" a virtual supermarket to purchase the items from their memorised shopping list. This gamified task also provided opportunities for users to increase the difficulty of this task, subject to satisfactory performance on previous levels. [134] similarly implemented a gamified cognitive stimulation programme in dementia care environments, focusing on structured task interaction rather than immersive delivery. More recently, [107] integrated gamified tasks within an immersive VR environment for persons with MCI, combining cognitive mini-games with meditation and physical tasks. Similarly, [43] used a gamified 3D multiple-object tracking (3D-MOT) task to train perceptual-cognitive skills in older adults with MCI and mild dementia. [113] applied interactive, game-like executive and processing speed challenges to individuals with MCI and subjective cognitive decline, structuring progression through task-based adaptive modules.

With regards to novel desktop or laptop-based applications (DESK), [49] investigated the acceptability of a cognitive stimulation desktop program that was interacted with exclusively via voice input. Here, several cognitive stimulation activities, designed to stimulate memory and communication, were presented both visually on a desktop screen, as well as being read out by a computerised voice. Participants would respond to the task on the screen through their voice (for example, responding with the answer to a quiz question) rather than through traditional keyboard-and-mouse or touchscreen methods. Though this setup was controlled entirely through 'Wizard-of-Oz' (where, unknown to the participants, a research team member was controlling the computer's outputs), the authors found high levels of acceptability and no

adverse effects after the voice-based interactions. Additionally, [40] implemented a novel desktop-based cognitive stimulation programme, combining structured exercises and gamified tasks in both MCI and AD populations. Other desktop-based interventions retained more traditional rehabilitation paradigms: [62] and [65] delivered structured computerised cognitive rehabilitation programmes in individuals with mild or early-stage dementia, while [77] and [65] implemented desktop-based task modules targeting executive control and memory processes in MCI subtypes. [127] implemented computerised cognitive training (RehaCom) in conjunction with high-frequency rTMS in PwMCI, across 10 supervised sessions, reporting significantly greater improvements in the combined intervention group, relative to stimulation alone. Across these studies, the desktop modality functioned primarily as a platform for structured cognitive task administration within controlled settings, distinguishing it from the more immersive (VR) or portable (APP/WEB) technologies.

Thirty-two of the 114 studies in this review employed a combination of technologies in their interventions. For example, [127] conducted an RCT-feasibility study using the individual Cognitive Stimulation Therapy (iCST) touchscreen tablet app (APP) in a one-to-one home-based intervention for PwD under the guidance of caregivers. The final prototype tested as part of this study also incorporated gamified ("game-like", GT) interactive features such as audio-visual stimuli and themed discussion questions, to be completed together by PwD and their caregivers. Gamified tasks were often combined with virtual reality, such as the virtual supermarket task in [143], or cognitive exergaming paired with VR tasks in [110]. In the large 2-stage SMART RCT by [138], a mobile CT platform was paired with dual-task Tai Chi delivered through either VR or offline instruction, showcasing a hybrid combination of APP, VR, and physical training modalities. In cases of potential ambiguity regarding the type of technology, we assigned terms consistent with those used by the article authors as opposed to reinterpreting them.

Only four articles that met our inclusion criteria used (social) robots in non-pharmacological interventions for PwD and PwMCI. One robot study [96] concerned the use of the "PARO" seal robot in a pilot RCT designed to evaluate whether the robot could be used for social interaction/stimulation intervention and reduce negative affect in persons with dementia. Two other robot studies concerned the use of wheeled humanoid robots. The study by [93] used a socially assistive tabletop robot ("Bomy") as a home-based platform for cognitive training in people with MCI. In this pilot RCT, the robot delivered five tablet-based cognitive-training programs, monitored user engagement, and provided verbal prompts. The robot-mediated intervention achieved high adherence and demonstrated the feasibility of robot-supported cognitive training in the home environment. [102] used the humanoid robot "Pepper" as an interface for a tablet through which familiar songs were played (designed to stimulate facets of cognition such as retrograde and anterograde memory). Outcomes for MCI persons were then compared against a group who only received the tablet as a means for evaluating to what extent a humanoid robot might provide an engaging tool for delivering psychosocial interventions. Whereas [102] evaluated whether a wheeled humanoid robot as an additional embodied interface could facilitate task involvement for the persons with MCI in the study, the usability study of [120] focused on user-centred design of a humanoid robot interface itself (of the "Kompai" humanoid robot). The researchers found that an "incremental design" method was effective for engaging persons with MCI in the user design task of the robot.

Whereas few robotics papers met our inclusion/exclusion criteria, many more papers that focused on virtual reality (VR) as a form of novel digital technology were extracted. VR interventions commonly combined cognitive tasks with physical activity ("exergaming") within immersive or semi-immersive virtual environments. For example, [78], utilised the VRADA (VR Exercise App for dementia and Alzheimer's Patients) with both PwD and PwMCI. The researchers emphasise the benefits of the sense of immersion that participants have when training in realistic, multisensory virtual reality environments. The [87] trial likewise combined physical exercise and cognitive stimulation during interactive cycling in virtual environments to support frailty and functional outcomes among PwD. Some interventions entailed participants carrying out VR-based cognitive training and exergaming independently (e.g., [97]. Others required the cognitive tasks to be carried out while exergaming, e.g., completing numerical calculations whilst cycling (see [78]). Another common use of VR is for visuospatial cognition training. In [117], for example, participants with amnestic MCI were required to navigate in a virtual environment towards different gemstones placed in particular locations and to reorient themselves to the starting position

thereafter. In a related study by the same author, [118] further examined the neural correlates of VR-based spatial cognitive training, incorporating measures of hippocampal and prefrontal cortical activity following the intervention. In a related immersive approach, [108] employed VR to deliver structured spatial and executive training tasks within a laboratory setting, reinforcing the use of VR as an embodied navigation-based training platform in MCI populations. In [75], persons with dementia of Alzheimer's type were trained in a virtual environment that required them to select among different navigation routes in order to reach real-world relevant destinations such as "home" and "shop". Similarly, [141] used VR with persons with dementia, requiring participants to navigate and interact with virtual environments using a head-mounted display, with instruction and monitoring provided in a supervised research setting. Bridging conventional computer-assisted training and fully immersive VR systems, [119] evaluated a mixed reality-based cognitive training platform in individuals with MCI, comparing MR-delivered spatial and executive tasks against traditional desktop-based cognitive training. The MR system overlaid digital elements onto the physical environment, representing an intermediate modality between screen-based cognitive rehabilitation and immersive VR interventions. Several newer studies combined VR, ecological simulation, and structured cognitive training. For example, [90] used tablet-based ecological VR (VESPA 2.0), while [110] used immersive VR integrating memory, attention, and physical-balance tasks.

In terms of trends in the types of novel digital technologies being deployed in cognitive and psychosocial interventions (see Fig 3), gamified tasks (GT) and mobile or tablet-based applications (APP) show the most sustained and consistent use across the full review period, with studies incorporating these technologies nearly every year since 2012. In contrast, the number of interventions making use of desktop or laptop-based applications (DESK) has gradually declined since 2021. Given that DESK-based cognitive training platforms have been commercially available for several decades, this decreasing trend may reflect the technological maturation of such systems and a corresponding shift in research focus away from desktop-delivered innovations toward more portable, immersive, or ecologically rich modalities.

Regarding the "newer" technologies, virtual reality (VR) stands out as a rapidly expanding area. All VR-based interventions included in this review were published after 2019, and the total number of VR studies (n = 27) is now comparable to that of APP and GT interventions, suggesting a strong and growing interest in immersive, interactive environments for cognitive, physical, or combined cognitive–motor training in PwD and PwMCI. This acceleration likely reflects both technological advances in affordable head-mounted displays and increasing evidence that VR can simulate meaningful real-world activities in an engaging and motivating format. Although robot-assisted interventions remain comparatively rare (only four studies in this review employed a social or humanoid robot) the studies span a range of robot types (e.g., PARO, Pepper, Kompai, and Bomy) and delivery contexts (clinic- or home-based). However, given the small number of studies and their distribution across the past decade, no clear temporal trend in uptake can be identified. Early work focused on social or emotional support (e.g., PARO), whereas more recent studies have used robots as delivery interfaces for cognitive training (e.g., [93]), possibly reflecting a gradual shift in conceptual focus rather than a quantitative increase in publications.

We define the implementation environment as the primary location(s) in which the intervention was deployed/provided to end users, as reported by the methods of each respective article. We label these environments using one of five categories: healthcare facility (HF), nursing home (NH), personal home (H), or research/academic facility (LAB). A fifth (NS) refers to "not specified", when the information could not be deduced from the article. That was the case in 10 of the 114 (8.7%) of the studies included in this review.

For the purposes of this review, a healthcare facility (HF) was defined as a location that is (a) not a participant's permanent place of residence, but rather an outpatient facility where a participant receives care or treatment (e.g., memory clinic, day care senior centre, rehabilitation centre, hospital). Explicitly, a memory clinic setting was reported in [111] and [58], a psychiatric outpatient centre setting in [71], a rehabilitation clinic setting in [32], and medical centre setting in [42]. Across all included studies, 43 instances of HF-based implementation were identified, making healthcare facilities the most common delivery environment in this review.

Research facilities (LAB) were defined as settings which were, for the purposes of the respective studies, solely used for data collection purposes. These facilities range from academic (university) research facilities, or research centres [42], and also include neutral locations that were not in a strictly academic location. In [38], this referred to participants' local parishes. 25 (22%) of the studies were solely conducted in a research facility, with a further four studies utilising both a research facility in combination with (at least) one other location: including personal homes (1) and healthcare facilities (3).

We define Nursing homes (NH) as public or private residential facilities for long-term care of persons unable to care for themselves. These types of facilities also typically have nurse assistants or carers available 24/7, and/or registered nurses, registered occupational therapists, registered physiotherapists and physicians available if needed. Nineteen (16.5%) studies were conducted in a nursing home facility, four of which employed other locations as well such as a nursing home and other healthcare facilities [32,57,60,86] as well as the participants' home in [38].

The Home environment (H) refers to the private residence of the participant. Of the 114 studies, 26 (22.8%) included interventions that were solely implemented at participants' homes. A further six studies used a combination of the participant's home and a medical research facility [83], a daycare centre [96], a healthcare facility [103,104], or across a range of locations [38].

It is important to note that only approximately one in four (288%, n = 32) studies included an intervention implemented/deployed in a home environment. However, this pattern was not evenly distributed across technologies. Notably, only two VR-based interventions were implemented in the home (see Fig 4). For robot technologies, two studies of the four studies [93,96] were conducted in the participants' homes.

## Summary of the available evidence for the potential effectiveness of the different technologies in the context of the interventions for improving cognitive, functional or well-being outcomes

From the 114 papers included in the review, 28 included a feasibility/usability study which often did not include cognitive, well-being or functional measures but users' experience measures (not reported here). From the 86 papers that included a study testing the effectiveness of the digital technology-based intervention, the majority (75.6%, n = 65) employed an RCT design which is the *gold standard*. The remaining employed a wide range of designs including cohort design, single intervention group (pre- vs post-intervention comparison) to non-randomised experimental controlled design. The sample size in the studies testing effectiveness also varied greatly, with the smallest sample size being 2 participants using a case study methodology in [62] and the biggest being 544 participants in [128].

To qualitatively summarise the evidence for assessing the potential effectiveness of the technologies used within the context of the intervention, we visually represented the outcomes findings (significant improvement = upward arrow; no effect of the intervention = dash; or significantly worse outcomes at post-intervention = downward arrow in the S3 Table) for all the intervention studies that reported statistical results for either cognitive, functional or well-being outcomes. Since we were particularly interested in cognitive outcomes, we further divided cognitive outcomes into the main cognitive sub-domains (e.g., attention, memory, working memory, executive function, language, reasoning, visuospatial skills, speed of processing). When specified, we followed the classifications of the specific measures into sub-domain constructs provided by the authors. When this was not explicitly specified we followed the most agreed upon classification in the relevant literature. Functional outcomes included objective or self-reported (by the end-users or caregivers) measures. Most of the studies that reported functional outcomes included measures of instrumental activities of daily living (e.g., in [6]) or other indexes of everyday functioning (e.g., in [114]); for example, caregiver burden and independence in [35], and physical fitness in [38]. Finally, the well-being category was defined broadly including measures of quality of life in [128], quality of relationships between patient and carer in [80] or other clinical symptoms that could impact quality of life. The majority of the 114 studies included in this review (79.8%, n = 91) reported cognitive outcomes; 29 (25.4%) reported functional outcomes, and 43 studies (37.7%) reported well-being outcomes.

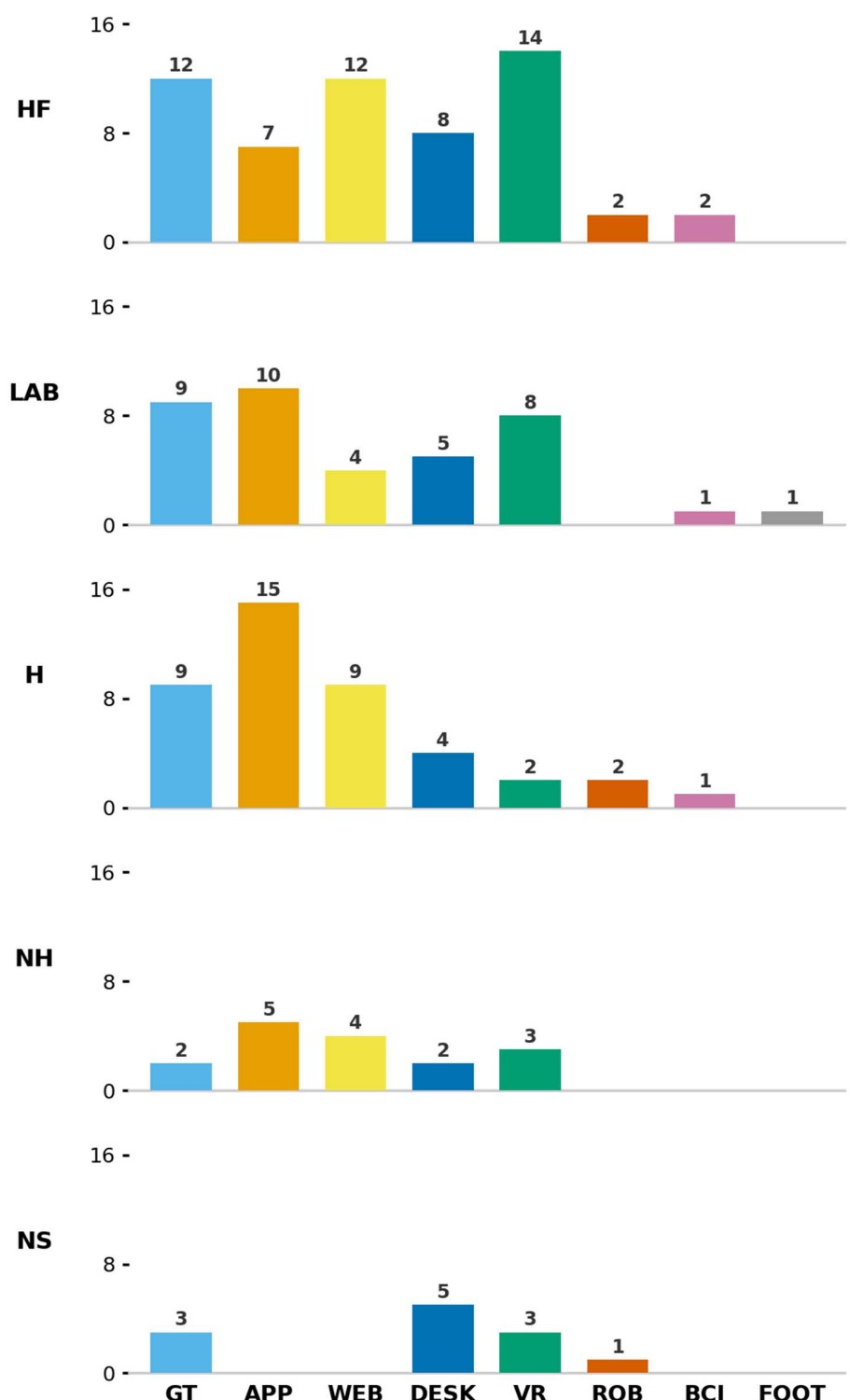

**Fig 4. Type of technology used with respect to the implementation environment study location.** HF = Healthcare facility; LAB = Research laboratory or other research setting; H = Home (Personal). NH = Nursing home. NS = Not Specified; 13 of the 117 studies included in this review were conducted in over multiple environments study settings.

Of the 91 studies that reported cognitive outcome, less than half (n = 34) reported that the technology-based intervention was effective in improving all of the cognitive outcome measures (e.g., [85,112,115]), with 17 studies reporting improvements in the majority of measures (e.g., [45,58,82]). There were a considerable number of studies (n = 40) that reported non-significant changes with the technology-based intervention, either in all (e.g., [48,76], the majority of (e.g., [92,123,129]) or in half (e.g., [127]) of the cognitive outcome measures. Some studies reported mixed results across different sub-domains. For example, [33] reported significant improvements in global cognition, but a significant worsening in executive function. Similarly, [64] reported improvements in working memory, memory, attention, and gnosis sub-domains, but reported significant worsening in global cognition, executive function, orientation and decision-making post-intervention. In a few studies, the effectiveness of the technology-based intervention was modulated by other factors such as social interaction (significant effects were reported only in the intervention group with high social interaction, e.g., [124]) or level of adherence to the protocol (significant effects only in the high adherence intervention group, e.g., [123]).

From the 43 studies that reported wellbeing outcomes, 19 found significant improvements after the digital intervention in all or the majority of wellbeing outcomes (e.g., [37,56,58,144]). However, almost half of the studies (n = 18) reported no significant change in all or the majority of wellbeing outcomes after the digital intervention (e.g., [38,48,58,66]). Finally, two studies reported mixed results: [124], reported improvement in 3 outcomes but worsening in one, and [123] reported no significant changes in two of the outcomes but worsening after the intervention in two of the outcomes.

Of the 29 studies that reported functional outcomes, only 13 reported significant improvements in all or the majority of the outcomes after the digital intervention (e.g., [35,46,51]). On the other hand, 16 studies reported no significant changes in the majority or all the functional outcomes after the digital intervention (e.g., [44,47,76]). Finally, one study reported a significant worsening in functional outcomes after the digital intervention [59] and one study [99] showed mixed results, with significant improvements in 5 outcomes but worsening in one of the outcomes after the digital intervention.

Despite the importance of adherence for the intervention's effectiveness, only 36.8% (n = 42) of the studies reported adherence (number of participants that complied with the prescribed number/duration of sessions). For those studies reporting adherence, it ranged from 11% to 100%, with three studies showing the lowest adherence: two in Home environments (e.g., 11% in [6], and 41% in [123]; see Table 1) and one in a Nursing Home environment (15% in [114]). Twenty studies reported a 100% adherence to the intervention; with the majority of studies being implemented in environments where there is supervision by a researcher/healthcare professional such as healthcare facilities environments, Laboratory environments and Nursing home environments. Only four studies [34,41,76,80] reported a 100% adherence in a Home environment.

## Discussion

The aim of the present scoping review was to summarise the research evidence on digital cognitive and psychosocial interventions in PwD and PwMCI. In line with [15] we observed that the number of publications per year has been increasing since 2012 with a peak in 2021. This could reflect policy-making efforts on digital healthcare, and the ongoing worldwide digital transformation, which has been further accelerated by the COVID-19 pandemic.

### What forms of novel digital technologies approaches are being used in cognitive and psychosocial interventions for persons with MCI and dementia?

The use of novel mobile and tablet applications and gamification or game-like elements has maintained a steady usage in cognitive interventions over the years covered by this review, with a peak of seven studies using GT and 6 using APP in 2024. This steady trajectory is owed, in part, due to its combination with other technologies. For example, the combination with virtual reality has resulted in the development of novel interventions including exergames (e.g., [78]) or more ecologically-valid memory games in immersive environments [143]. Considering recent systematic reviews showing the benefits of gamification in improving other health-related outcomes [146,147], it could be argued that the incorporation of

gamification in cognitive interventions may continue to be a growing trend in the near future. The use of gamification for adapting and personalising cognitive training (e.g., as seen in [143]) may also continue to be a valuable component contributing to engagement and potentially long-term adherence to such interventions.

The relative lack of articles describing robotics-based cognitive or psychosocial interventions was surprising. [15], for example, extracted 48 articles in their scoping review on digital healthcare in relation to dementia and cognitively impaired persons. It can, however, be pointed out that our exclusion criteria and research question(s) were somewhat more restrictive than those of [15] in relation both to: i) therapeutic approach, and ii) target group. In relation to i), the exploration of digital technology usage was not restricted to interventions and the scoping review included articles reporting digital technology usage in diagnostics as well as acceptability and non-cognitive focused studies, including observational studies (only 8 focused on cognitive outcomes). In relation to ii), cognitive impairment was not restricted to MCI and dementia and included articles whose investigations additionally focused on mental and physical health among other forms of impairments.

A recent systematic review on the use of robotics in cognitive interventions for PwMCI and PwD from [24] extracted 19 intervention-based articles. However, the search string did not exclude cognitive impairments other than dementia or MCI (including "cognitive impairment", and "senescent forgetting" among other terms) thereby broadening their scope. Our study required "*a diagnosis of dementia or MCI according to international standards/criteria*" and was very specific in terms of cognitive impairment. Five of 19 articles in [24], used the PARO robot but either in groups or in individual sessions, and would not necessarily have qualified as cognitive or psychosocial interventions according to standard definitions. The use of robotics in interventions is more commonly associated with well-being, dyadic social-based interventions, or assisting participants in cognitive interventions rather than as a platform for the delivery of the cognitive training or intervention in itself. This argument is somewhat consistent with [15] who found that robotics used in digital healthcare was disproportionately low for cognitive interventions.

Although it has been suggested that social robots in particular could increase the autonomy and independence of persons with dementia (see [148]), there may be several factors that explain the relatively low number of studies investigating the effectiveness of robot-based cognitive or psychosocial interventions and delivery methods for cognitive training, in people with MCI or dementia. Besides the obvious limitation of the high cost of the physical hardware, other issues relating to acceptability should be considered. For example, [149] pointed out that socio-cultural attitudes toward robots along with the difficulties relating to the technological aspects (e.g., failure to recognise speech) may result in less autonomy for end users and a greater burden to caregivers. Findings regarding the acceptability of social robots in persons with cognitive impairments are mixed and vary depending on the severity of impairments and the socio-cultural context [117,150].

In summary, whilst it has been suggested that robotics is an emerging and novel technology in relation to different forms of interventions evaluating cognitive impairments (including dementia and MCI), its use in cognitive and psychosocial interventions (particularly involving cognitive training) at a level that meets our relatively strict exclusion/inclusion criteria for quality and diagnostic criteria, does not appear to have evolved much over the scoping period reviewed. Although, a substantial number of papers that included robots were captured by our search, most of these papers were rejected because of low quality (e.g., the criteria for the diagnosis or the diagnosis of MCI/dementia were not explicitly described) or because they presented the technology at very early stages of development or technology readiness (e.g., design or concept paper or acceptability study).

In terms of other novel technological approaches, we observed that all the studies including VR-based interventions were published after 2019. Thus, our scoping review suggests that VR is an emerging technology for cognitive and cognitive-physical interventions in PwD and PwMCI. This trend could be partially due to the greater accessibility in terms of the cost of VR relative to social robots, which could facilitate uptake in both the general population and healthcare practice [151,152]. The increasing trend of VR use identified here could also be related to the type of end-user experiences enabled by this technology. It has been pointed out that VR could offer more ecologically valid interventions by allowing

more realistic three-dimensional environments and interactions [153]. However, our review suggests that this technology has predominantly been used in particular environments and patient populations as discussed in the section below.

**What is the geographical mapping of the use of the technologies? What are the gaps in the development/ research/implementation of technologies in Europe?**

We find that novel mobile and tablet applications, as well as gamified tasks, have been studied at least once in the majority of the countries in our review. Specifically, 19 of 29 countries returned at least one instance of using mobile or tablet applications; whereas Gamification or games-based interventions (GT) were used at least once in 21 of the 29 countries.

There does not appear to be any clear global trends related to the use of this APP and GT in non-pharmacological interventions; however, our review found that 17 of the 28 studies using VR came from countries in Asia: South Korea, China and Taiwan. This finding supports [154] who similarly noted considerable studies using VR (and, in particular, VR exergaming) emerging from these countries. Considering that VR has seen a steady number of studies since 2019 (Fig 3), we can consider that these early advancements in novel virtual reality-based interventions are being driven (in large part) by the studies being conducted in Asia. Perhaps surprisingly, many of the seven countries that accounted for the remaining VR studies (Greece, Estonia, Poland, Portugal, Italy, Netherlands and Mexico) were amongst the European countries with the lowest GDP in our study (relative to Germany, Italy, Spain, France, UK). The use of VR in these (somewhat) lower GDP countries may emphasise that this technology does not come with the same cost-related limitations as others, whilst being widely accessible as compared to other emerging technologies (such as robots).

The remainder of the studies from Europe came mostly from Western European countries: Italy, Spain, France, and Germany. See S2 Table for breakdown of technology per geographical location.

Although many countries are represented by the studies in our review, it is particularly noteworthy that there were no studies from the continents of Africa, Central, or South America (with the exception of Mexico, see Table 4). This observation highlights, at a very high level, imbalances and inequalities in the development or implementation of digital technologies globally. However, it is beyond the scope of this review to analyse the specific and complex reasons for this geographical gap. In their recent review, [155] alluded to specific cultural and/or religious barriers in Africa that may propose challenges to the implementation of traditional (and, by extension, digitalised) cognitive interventions for persons with dementia. Similarly, [156] point out that although African countries account for almost one-quarter of the global burden of disease, they have less than 1% of the world's financial resources and only 3% of trained health workers. For countries in Latin (including Central and South) America, several authors refer to factors including: a lack of dementia awareness in the general population; social stigma; a lack of timely diagnosis and intervention from the medical system (including budget constraints); aversion to technology-based interventions; political and economic instability; other cultural barriers [157–159].

Here, we do not claim that all of these issues are what explicitly led to a lack of studies from these countries in our scoping review, since our findings derived from our specific inclusion criteria. Instead, we use this opportunity to highlight that technology-based interventions and development will (amongst many other factors) be impacted by existing global (social, political, infrastructural, and economic) disparities. These disparities risk being further perpetuated with focuses on particular technologies (e.g., VR or web-based) and for/with certain populations (e.g., Western Europe) with existing technological and/or social infrastructure to facilitate their development and uptake. Thus, in addressing global health challenges such as dementia, digital technology health interventions – largely being driven by high-income countries (Table 4) – should also consider how such interventions can be accessed and utilised more globally, and specifically in lower and middle-income countries (i.e., across Eastern Europe, Africa, South America, and parts of Asia) that are currently not receiving the same amount of attention. One possible approach for addressing gaps would be for more affluent nations that dedicate resources to Digital Health applications, e.g., in Europe and North America, to provide opportunities for bilateral research with countries that lack such resources/development, e.g., in Africa and South America.

**Are particular technologies being used in specific environments or subgroups of patients?**

Several authors have suggested an increasing trend in the use of robots for health care in industrial applications to home and consumer-based markets [110,160]. This is not supported by the present scoping review, in the context of non-pharmacological interventions with PwD and MCI. This gap in home use may be related to the higher cost of these technologies and some other factors discussed above. Similarly, only two of the studies included in the present review, which used VR technology, implemented or tested the intervention in a home environment. Concerning the target populations (PwMCI and PwD), some technologies appear to have been implemented or tested predominantly with one group of patients (see Table 3). This is the case for VR, for which the majority of studies were with PwMCI. There were also more studies using gamified tasks with PwMCI than with PwD. However, applications (APP) appear to be more widely studied with PwD than PwMCI.

One of the reasons why the majority of studies with VR interventions were investigated with MCI may be the potential for negative experiences or unintended harm when using this technology. Although more recent reviews suggest that "cybersickness" is relatively low in immersive VR with people with neurocognitive disorders [161,162], and that there is good acceptability in older persons with mild to moderate dementia [34,41,70], potential negative effects may deter researchers and clinicians from using immersive VR with the more vulnerable population of PwD. For example, authors have also noted the possibility of unintended harm (e.g., falling over, eye strain) when interacting with virtual objects in immersive environments [163,164], particularly with older adults who have sensory impairments such as reduced vision. Our review also suggests that VR may not yet be a feasible target technology for remote, home-based cognitive or psychosocial interventions. However, this may be due to the state of widespread usage of the technology itself, rather than its specific application within cognitive interventions. Put another way, unlike other technologies such as mobile phones or tablets, VR is still not well embedded in day-to-day social and personal life and thus requires additional technical and physical resources to implement (e.g., purchase of specific headsets, training of staff, dedicated physical spaces to use).

**What is the evidence for the potential effectiveness of the different digital technologies in the context of the interventions for improving cognitive, functional or well-being outcomes?**

In the present review, we aimed at qualitatively summarising the evidence on the potential effectiveness of the technology-based interventions, to inform future research. To do so, we visually summarised (see S3 Table) the effects of the intervention on cognitive, functional and wellbeing outcomes as reported in the studies included as an improvement, no change or worse outcome (pre- vs post-intervention or/and control vs intervention group comparison). Many of the studies (about half) did not report significant improvements in the majority or all of the cognitive outcomes. Similarly, many of the studies reporting functional and wellbeing outcomes did not find significant improvements in the majority or all of the measures. Although the majority of the studies employed an RCT methodology, there was great heterogeneity across studies in the type and number of outcomes reported, in the sample size and the design of intervention (e.g., dose) which could account for some of the variability in results.

Several meta-analyses of the effectiveness of VR interventions in MCI and dementia have been conducted in the last five years [153,165,166] and overall, they report significant effects on cognitive outcomes in people with MCI. There are nevertheless mixed results from these studies regarding effects in specific cognitive domains. The most recent meta-analysis that included 20 studies [153] concluded that VR cognitive training in MCI significantly improved global cognition outcomes and the subdomains of memory, attention and construction/motor performance but not executive function and language. The authors [153] also concluded that VR training (4 RCT studies) significantly improved global cognition, memory and executive function in PwD. As pointed out by [88], the literature examining the effects of VR interventions in populations with cognitive impairments is in its infancy. There is high heterogeneity in the results [153], as well as the methodologies employed (e.g., dose, type of VR technology, type of training, and sample sizes) as also suggested in the present review.

In addition to those described above, several other meta-analyses have been conducted in the last five years to examine the effectiveness of gaming-based interventions [154,166–169]. Results from this literature are also mixed, and there are different definitions and approaches regarding the use of gamification or game-like tasks in interventions for people with cognitive impairments. For example, [166] conducted a meta-analysis study of 14 RCT studies using serious game digital technology in MCI and dementia, without restrictions in terms of type of intervention (e.g., exergames, video games, cognitive training, biofeedback), and reported significant benefits on cognitive and daily behavioural ability and depression. [167] meta-analysis study employed a narrower definition of gaming technology and included only studies that used brain gaming. According to the authors, the defining characteristic of brain gaming technology is the ability to adapt games based on the level of difficulty, providing the end user with a challenging and competitive experience. In their meta-analysis of 14 RCT studies, the authors concluded that brain gaming interventions were not effective, as compared to a control intervention, in improving cognitive outcomes in PwMCI and PwD. Both meta-analysis studies [153,167] also pointed to high heterogeneity in the design of studies using gaming interventions such as dose, type of training, sample size, and severity of impairment. In our review we did not restrict our scope to the type of gaming technology used in the intervention, so further research is needed to understand the potential of specific types of gaming technology for intervention in MCI and dementia.

As discussed, research on cognitive or psychosocial interventions using robots to deliver the intervention, and not just as companions, is at its infancy. There is consequently a scarcity of systematic reviews and/or meta-analysis studies investigating effectiveness on cognitive, functional or well-being outcomes. The four papers with robot-based intervention included in this review provides insights into how this technology may need to be utilised, through a focus on social interactive factors and design stages, for it to realize its full potential for cognitive or psychosocial interventions. Three of the studies included here tested robot interventions with PwMCI [93,102,120], one reported improvement in the majority of cognitive outcomes [102], another in one of the three cognitive outcomes, while the other was a feasibility study and did not include cognitive outcomes. The four study [96] tested a robot intervention in PwD and did not report significant improvements in any of the cognitive, functional or well-being outcomes.

Finally, for the other technologies (APP, DESK and WEB) that have been employed for cognitive and psychosocial interventions for a longer time, there have been many systematic reviews and meta-analyses conducted in the last decade. In one of the most recent meta-analysis studies, [170] concluded that CCT had significant improvements in verbal episodic memory, visual episodic memory, and working memory in PwD. However, when they compared supervised vs unsupervised (home setting) CCT, they found that supervised CCT showed the greatest benefits. They also concluded that CCT overall had significant improvements on verbal episodic memory, but not on visual and working memory, in PwD. The studies included in [154] meta-analysis employed mostly desktop computers (74.3%), touch screen computers or tablets (20%), while a small percentage (5.7%) of the studies used immersive VR. It is important to note here that these technologies APP/WEB/DESK may not be well represented in our review with regard to effectiveness due to our inclusion criteria based on the novelty of the technologies. As already mentioned these technologies have been developed and used for a longer time, and so they were more likely to be excluded based on novelty of the digital technology or their application. Despite this limitation, as discussed earlier, only a small number of studies that included APP, WEB or DESK technologies were tested in an unsupervised (H) environment.

## Limitations

One of the limitations of our scoping review is that we excluded a substantial number of studies that reported digital cognitive or psychosocial interventions with the target populations that did not use novel technology or development or application. The rationale was that we would include studies that spoke to emerging and current trends in technology development, use and implementation in terms of improving the cognitive, functional and well-being status of people with cognitive impairments. We defined "novelty" in terms of digital/technological interventions that have not been already

commercialised and have been in the healthcare market for some time. We included already commercialised digital interventions only if they reported in the study a novel digital or application component that aimed at enhancing the potential effects of the intervention. Therefore, the trends observed, and gaps identified do not concern all the digital interventions but only those ones that were judged as having a novel component according to our definition.

There was also high heterogeneity in the design of the interventions (e.g., sample size, length and dose of the intervention), and the reporting of effects on cognitive, well-being and functional outcomes. For example, while some studies included an active control comparison group, others included a passive control group.

Another limitation related to the definition of technology type, and the analyses comparing different types of technologies. This was particularly the case for the *Game (GT)* category. Game digital technology in the context of non-pharmacological interventions may take many forms such as cognitive tasks that include some gamified aspects, exergaming activities that combine cognitive with physical training, video games and biofeedback games [169]. Other authors (e.g., [167]) refer to a specific category of brain games which are designed with adaptive levels of difficulty so that they remain challenging and competitive. Similarly, VR technology is undergoing a rapid rate of development and cannot be considered a monolith [171]. For example, different types of headsets used in the studies in this review (e.g., the Samsung Odyssey [88], the Oculus Go [78] and HTC Vive [97] bring with them different levels of cost, comfort, immersion, and modes of interaction (i.e., some VR environments require the use of handheld controllers, whilst others can utilise hand-tracking). Such factors are likely to impact long-term acceptability and engagement with cognitive interventions being delivered through this technology. Furthermore, many of these technologies are being used in combination. For example, many exergames include exercise with games in VR settings [154]. In the present review, we argue that it is important to understand how different types of technologies are being used, in what context, populations and outcomes. In line with this, many meta-analysis studies have investigated the effectiveness of specific types of technologies in PwD and/or PwMCI (e.g., [153,165] for VR and [166] and [154] for games). However, this is not a straightforward task, and if the number of studies combining different technologies increases in the future, it may be difficult to study and compare different types of technologies for non-pharmacological interventions in people with cognitive impairments.

### Implications for research

The findings discussed above point towards several outstanding questions that could benefit from further research: i) more research is needed to investigate the effectiveness of specific technologies and intervention designs for improving outcomes in unsupervised (home) environments in PwMCI and PwD; ii) given that VR seems to be an emerging technology that is being increasingly developed and implemented (since 2019) to deliver non-pharmacological interventions particularly in PwMCI, further research is needed to investigate how this technology could be made more accessible and scalable to deploy interventions at home; iii) further research is needed to provide insights into how the different technologies, settings and individual characteristics may influence (and potentially enhance) adherence to cognitive and psychosocial interventions in PwMCI and PwD, and in particular it would be important to differentiate between attendance and duration adherence; iv) more research is needed to understand how robots could be used to deliver cognitive interventions, and what is the effectiveness of interventions delivered with robots in cognitive outcomes.

### Conclusions

One of the key findings is that VR is an emerging technology in relation to delivering mostly cognitive and physical (e.g., exergaming) interventions in PwMCI, but also PwD. Game and application-based interventions seem to have gained traction in the last decade. Our review also suggests that the evidence with regard to the effectiveness of the technology-based interventions in improving cognitive outcomes is heterogeneous, probably partially due to the methodological heterogeneity and variability across studies. There seems to be a present gap in the use of robots to deliver, as opposed to merely provide assistance in, cognitive intervention in PwMCI and PwD, which could be related to several

factors as discussed earlier. Overall, there seems to be an increasing trend since 2012 of published studies testing a wide range of novel digital technologies to deliver cognitive or psychosocial intervention in PwMCI and PwD. Despite the growing interest in digital healthcare solutions that could increase accessibility, scalability, and sustainability, our review supports that the majority of the digital interventions for cognitive and psychosocial interventions are being developed and tested for administration under real-time supervision by a trained professional and not for unsupervised home environments in spite of the potential for many of the aforementioned technologies to be applied for home use, and even mitigate issues concerning patient mobility (e.g., VR using exergaming).

Finally, we observed high heterogeneity in terms of intervention protocol (e.g., length and dose), the types of activities included in the intervention (e.g., real-life scenarios daily activities, cognitive tasks, physical activity, social interaction), the hardware (e.g., immersive vs non-immersive VR technology) and the type of outcomes included. Therefore, as the number of studies designing or testing non-pharmacological interventions using novel digital technologies continues to increase it is important to produce evidence-based guidelines for their implementation in healthcare settings.

## Supporting information

**S1 Table. Initial Search String with Database-specific Syntax Requirements.**
(PDF)

**S2 Table. Types of Technology Used Per Country and Per Region.**
(PDF)

**S3 Table. Outcome Summaries for Extracted Articles.**
(PDF)

**S4 Supplementary Material. Search Strategy.**
(PDF)

## Author contributions

**Conceptualization:** Ana B. Vivas, Imran Khan, Qarin Lood, Pierre Gander, Simon Nielsen, Robert Lowe.

**Formal analysis:** Ana B. Vivas, Imran Khan, Qarin Lood, Pierre Gander, Mia Dong, Simon Nielsen, Robert Lowe.

**Funding acquisition:** Robert Lowe.

**Investigation:** Ana B. Vivas, Imran Khan, Qarin Lood, Pierre Gander, Mia Dong, Simon Nielsen, Robert Lowe.

**Methodology:** Ana B. Vivas, Imran Khan, Qarin Lood, Pierre Gander, Simon Nielsen, Robert Lowe.

**Writing – original draft:** Ana B. Vivas, Imran Khan, Qarin Lood, Robert Lowe.

**Writing – review & editing:** Ana B. Vivas, Imran Khan, Robert Lowe.

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
