## [Decision Letter · Decision Letter 0]

25 Feb 2025

PONE-D-25-01664The Use of Digital Technology in Non-Pharmacological Cognitive and Psychosocial Interventions for People with Dementia and Mild Cognitive Impairment: A Scoping ReviewPLOS ONE

Dear Dr. Lowe,

Thank you for submitting your manuscript to PLOS ONE. After careful consideration, we feel that it has merit but does not fully meet PLOS ONE’s publication criteria as it currently stands. Therefore, we invite you to submit a revised version of the manuscript that addresses the points raised during the review process.

If applicable, we recommend that you deposit your laboratory protocols in protocols.io to enhance the reproducibility of your results. Protocols.io assigns your protocol its own identifier (DOI) so that it can be cited independently in the future. For instructions see: https://journals.plos.org/plosone/s/submission-guidelines#loc-laboratory-protocols. Additionally, PLOS ONE offers an option for publishing peer-reviewed Lab Protocol articles, which describe protocols hosted on protocols.io. Read more information on sharing protocols at . Additionally, PLOS ONE offers an option for publishing peer-reviewed Lab Protocol articles, which describe protocols hosted on protocols.io. Read more information on sharing protocols at https://plos.org/protocols?utm_medium=editorial-email&utm_source=authorletters&utm_campaign=protocols..

We look forward to receiving your revised manuscript.

Kind regards,

Mi-So Shim

Academic Editor

PLOS ONE

Journal Requirements:

2.Thank you for stating the following financial disclosure: [The authors received a budget from Forte (REF: 2021-01781) –  “Digitalized Non-pharmacological Interventions for People with Dementia: Reviewing Scandinavian and International Contexts”, (budget 996,300SEK) to carry out this scoping review research based on an expert panel-reviewed competitive grant application.].

Please state what role the funders took in the study. If the funders had no role, please state: "The funders had no role in study design, data collection and analysis, decision to publish, or preparation of the manuscript.

5. We note that Figure 6 in your submission contain [map/satellite] images which may be copyrighted. All PLOS content is published under the Creative Commons Attribution License (CC BY 4.0), which means that the manuscript, images, and Supporting Information files will be freely available online, and any third party is permitted to access, download, copy, distribute, and use these materials in any way, even commercially, with proper attribution. For these reasons, we cannot publish previously copyrighted maps or satellite images created using proprietary data, such as Google software (Google Maps, Street View, and Earth). For more information, see our copyright guidelines: http://journals.plos.org/plosone/s/licenses-and-copyright.

1. You may seek permission from the original copyright holder of Figure 6 to publish the content specifically under the CC BY 4.0 license.

In the figure caption of the copyrighted figure, please include the following text: “Reprinted from [ref] under a CC BY license, with permission from [name of publisher], original copyright [original copyright year].

6. Please include captions for your Supporting Information files at the end of your manuscript, and update any in-text citations to match accordingly. Please see our Supporting Information guidelines for more information: http://journals.plos.org/plosone/s/supporting-information..

Reviewers' comments:

Reviewer's Responses to Questions

**Comments to the Author**

1. Is the manuscript technically sound, and do the data support the conclusions?

Reviewer #1: Partly

Reviewer #2: No

Reviewer #3: Partly

2. Has the statistical analysis been performed appropriately and rigorously? 

Reviewer #1: N/A

Reviewer #2: No

Reviewer #3: N/A

3. Have the authors made all data underlying the findings in their manuscript fully available?

Reviewer #1: No

Reviewer #2: No

Reviewer #3: Yes

4. Is the manuscript presented in an intelligible fashion and written in standard English?

Reviewer #1: Yes

Reviewer #2: No

Reviewer #3: No

5. Review Comments to the Author

Reviewer #1: 1Overview:

I am grateful for the opportunity to review this manuscript, which presents a scoping review on digital health technology-enhanced cognitive and psychosocial interventions in neurocognitive disorders. The review addresses relevant research gaps, which have been well elaborated on in the Introduction. I’d particularly like to commend the authors for committing to completing such a high volume of work. The results are relevant and useful, highlighting that there are still vast opportunities for improvement for unsupervised home-based solutions to optimize healthcare resource allocations in the treatment and management of neurocognitive disorders with the help of digital health technologies.

However, I have concerns that the search strategy might not have been sensitive enough to capture all relevant literature and the methodological choice in the screening of records has resulted in an incomplete picture of the current stage of evidence. The search strategy and screening process are the most critical elements in ensuring the validity of the results of such a review. Therefore, I provide below detailed recommendations for improvements to these and additional aspects of this work to strengthen its completeness, robustness, and future impact in the field.

2Major Comments:

1.Section “Search” (page 13): It is laudable that the authors consulted six different libraries, supporting the comprehensiveness of their approach. However, compared to previous reviews, the search strategy seems rather incomprehensive, and some key terms may be missing. Such search terms may include but are not limited to exergaming (MeSH term and all related subterms), X-box, video games, game-based, gamification, gamified, smartphone, television, interactive computer game for the intervention, and mild neurocognitive disorder and related terms for the population. As a result, the sensitivity of the search strategy in identifying all relevant literature is potentially insufficient. This impression seems to be supported by the low number of identified records for such a broad review compared to similar reviews in the field and has also been acknowledged in part by the authors in the Discussion (lines 667-684 that explain this observation with a comparison of eligibility criteria, rather than the sensitivity of the search strategy).

Good practice would be to pilot test the search strategy to ensure that it identifies known key publications that are consistent with the research question and eligibility criteria (see the Cochrane Handbook for Systematic Reviews of Interventions; https://training.cochrane.org/handbook). I recommend clarifying whether this recommendation has been followed and what the outcome of this process was (e.g., was the search strategy deemed sensitive enough and/or was the search strategy refined?). I strongly recommend to clarify how the search strategy developed, to consult a librarian to update the search strategy following Cochrane Handbook, and transparently report on the results from the search of each library.

2.Classification of Exergame Technology: The manuscript would benefit from clarifying the classification and interpretation of exergame-based interventions. According to the aim and eligibility criteria, this review concerned cognitive and psychosocial interventions. However, the results report on exergame as a “physical training”, which appears to be outside the scope of this review. Nevertheless, since exergame-based training typically targets physical and/or motor-cognitive training, I agree with the authors that this intervention modality should be within the scope of this review. Therefore, I recommend updating the search strategy to specifically include exergames, clarifying how exergame-based interventions were categorized, and reconsidering (or explaining why you did not) the use a separate category for exergame technologies. Furthermore, the reporting of results should be aligned with the defined objectives (specifically include physical and/or motor-cognitive training modalities). I recommend consulting the following publications to derive methodological choices (particularly regarding the search strategy and categorization) from: https://doi.org/10.3389/fnagi.2022.859715,
https://preprints.jmir.org/preprint/65746,
https://doi.org//10.1002/14651858.CD013853.pub2,
https://doi.org/10.1016/j.arr.2024.102385, or https://doi.org/10.1038/s41746-024-01142-4).

3.Criteria for Technology Categories: The authors note that many studies implemented multiple different technology modalities. To enhance the reproducibility of this review, the manuscript would benefit from providing the criteria that were employed to define the eight categories of technologies analyzed and the specific criteria on how categorization was handled in cases where multiple categories matched.

4.Search results and selection: The manuscript would benefit from providing your results on the inter-rater agreement for all steps of this work that were performed by two independent reviewers (including screening and data extraction), for example by calculating Cohen's kappa, to make more transparent how good the agreement was in these procedures. Please also provide a rationale for the large number of reviewers in relation to the small number of records screened and discuss the limitations of your approach (potentially reduced inter-rater reliability due to the number of reviewers and data extractors involved) in the "Limitations" section.

5.Section “Synthesis” (Page 16 ff) as well as lines 529-538 and 597-604 describe methods of this review and should therefore be moved to the Methods section. In this regard, please also clarify how pilot, feasibility, and qualitative studies were handled in the exploration of effectiveness, given that such studies are not designed to be powered for robust evaluations of an intervention's effects. Furthermore, I recommend also reporting on the observed effect sizes and confidence intervals to provide insight into the clinical meaningfulness of these effects, as solely relying on p-values might not provide a complete picture of the evidence of efficacy given the inclusion of pilot, feasibility, and qualitative studies.

6.Focus on Technology Component: The manuscript puts a heavy focus on the technology part of an intervention. In my view, conceptual decisions and algorithmic decision trees in the training concept together with purpose-developed software build the heart of every digital health technology-enhanced training, whereas it is well-justified to rely on well-established and commercially available hardware as a means to an end to deliver the training to promote accessibility of the resulting intervention. This viewpoint aligns with the "intervention first" rather than a "product first" approach suggested by Torre and Temprado (https://doi.org/10.3389/fnagi.2022.859715). Therefore, I strongly recommend providing more nuance regarding the design and delivery of the interventions that were supported by the technology. Torre and Temprado’s paper provides good examples of how this could be achieved and I recommend using this or similar previous contributions as a groundwork to build on.

7.Limitations: The first limitation regarding the exclusion of all “non-novel” technologies used for interventions requires further explanations. From this limitation, it remains unclear which aspects (of software and/or hardware) of the technologies required novelty to be included in this review. As an example, the authors reported that the included studies used different types of VR headsets, including the Oculus Go or HTC Vive, both of which refer to off-the-shelf headsets. The same also goes for screens or other devices (e.g., exergame hardware like Wii Balance Board or Wii Remote) to deliver the intervention. Therefore, the manuscript requires clarification on which exact criteria led to the inclusion of the mentioned examples and particularly the exclusion of this large amount (88 papers) of potentially relevant studies.

In this regard, I strongly recommend rethinking this methodological choice. While the authors justified this methodological choice regarding exploring trends in research, none of the research questions were formulated accordingly and imply comprehensiveness by covering all existing relevant literature. This mismatch might be misleading to the readers and potentially leads to biased findings, particularly regarding the efficacy analyses. To better align the methodology with the existing research objectives, I recommend including all technology-enhanced interventions relevant to provide a comprehensive answer to the research questions, and addressing trends as part of a sub-analysis to provide further depth to the findings.

3Minor Comments:

8.I recommend providing a justification for conducting a scoping instead of a systematic review.

9.Context of VR Applications: The manuscript would benefit from clarifying whether the context of this study was limited to “immersive” VR applications or whether other forms of VR as well as AR and XR were also included. This clarification is especially important in the context of exergame-based training, where most research has focused on non-immersive VR applications like video games displayed on a screen and now more and more AR applications, since VR poses some risks (e.g., for falls) when exercises are performed in a standing position.

10.Implications for Research: The discussion would benefit from providing a dedicated section “Implications for Research” and providing clearer recommendations on how future research can benefit from and build on the findings obtained.

11.Handling of Insufficiently Reported Data: Please report on how the limitation “In relation to the quality of the studies, many studies did not report the information we needed to answer our research questions (e.g., implementation environment and adherence)” was handled. Were authors contacted (and how many times) to obtain the insufficiently reported data required in this work?

12.Several sections in the results are interpretative and therefore belong to the discussion section, including: “This may suggest that the majority of the available digital technologies for cognitive/psychosocial interventions with PwD and MCI may not be yet ready for remote individual home use.” (Page 21, lines 458 - 460) or “This could suggest that some technologies are being designed for a particular group of patients” (page 21, lines 473 - 474)

13.Page 24, lines 587 - 859: I recommend distinguishing between attendance and duration adherence, as these refer to different constructs.

14.Page 12, line 169: The manuscript would benefit from clarifying which criteria the judgement whether the record was based on a “prestigious/reputable conferences (e.g. IEEE-based) with extended papers”.

15.Page 12, line 170: The manuscript would benefit from providing a rationale for the 2012 limit.

Reviewer #2: The novel digital technology interventions in the article for older adults with cognitive impairment and neurodegenerative diseases have some relevance. However, there are still some issues in the article that need to be revised. Mild cognitive impairment and dementia should be discussed separately, and the current definition of “digital technology” is broad, so it is recommended that the classification be refined by function (e.g., diagnosis, intervention, monitoring) or by type of technology (e.g., AI, VR, wearable devices). The most recent research should be included. Most cases only describe the technological framework and lack controlled trials or long-term follow-up data to support efficacy. Search strategies, inclusion/exclusion criteria, and quality assessment methods for scoping reviews need to be clarified to avoid selection bias. Some terms are used inconsistently, and reference to international consensus is recommended.

Reviewer #3: Dr. Suad Ghaben

PONE-25-01664

The Use of Digital Technology in Non-Pharmacological Cognitive and Psychosocial Interventions for People with Dementia and Mild Cognitive Impairment: A Scoping Review.

Dear Authors

Please find my comments on your manuscript below. Your manuscript is a scoping review on the Use of Digital Technology in Non-Pharmacological Cognitive and Psychosocial Interventions for People with Dementia and Mild Cognitive Impairment. Your results highlighted eight main categories of digital technologies that has been utilized within the study timeframe mainly gamified technologies, mobile app, web-based approaches, and virtual reality. Positive cognitive outcome was the most detected outcome compared to functional and well-being outcomes.

I will start by commenting on the manuscript parts in order highlighting major and minor revisions.

Title:

Short, concise, and reflective. No comments

Abstract:

Major:

•Add to the methods\design section how screening was performed.

•Add to the methods section “We followed the Joanna Briggs Institute guidelines and the framework”.

•Add to the results section the number of studies you analyzed.

Minor:

•Use objective instead of background to align with the journal style.

•No need to mention the results of the geographic data in the results section.

•Replace “our study” with “this study” in the results and discussion sections

Introduction:

Major:

•Suggest to add subheading “Literature review”

•98-99: cite

•100-102: cite

•Lines 84-85: suggest to search literature on bibliometric studies on the topic and cite accordingly.

•Lines 117-128: be specific in identifying the research gap.

Minor:

•Line 55: PWD: spell out in full on first mention

•Line 103: deployment/implementation: this style of writing is not accepted in academia. Rather you can use one term or describe each term separately.

Methods:

Major:

•Rearrange the methods section and add the mentioned subheadings in order: search strategy, eligibility criteria, study selection, data extraction and synthesis.

•The methods section lacks the main description of the followed methodology, which is mostly described under the results section.

Minor:

•Lines 224-225: this is consistent with the findings … move this interpretation to the discussion section. Result section is to present your results only.

Results:

Major:

•rearrange the results section

•add the following subheadings: Study methods, study characteristics, characteristics of digital interventions, effectiveness outcomes

•lines 133-235 : data charting process ”Four team members … consistency of the items extracted. Move this part to the methods section under the data extraction and synthesis subheading .. and under the “Study characteristics” subheading under the “Results” section describe the end user population, geographic location and funding sources .. etc.

•lines 288-289, 417-419, 483-484, 518-519: delete the RQs and describe the answers for each question in separate paragraphs.

•275-286: can you justify the deployment of the PICO framework? the PICO framework was not included as an eligibility criterion.

•Any description related to the applied farmwork should be moved to the Methods section.

Minor:

•keep consistency of the methods and results sections

Discussion:

Major:

•lines 640-652: no need to describe the research gap here. Rather you need to go straight summarizing your main results.

•Line 898: Other gaps in knowledge… the discussion section is to interpret your results and compare it with existing literature... rather than discussing the knowledge gap. It can be accepted to discuss the knowledge gap only if your goal is to find the research gap in digital technologies ….. etc.

Minor:

•Lines 944-946: what are the criteria of considering the digital interventions as novel? This was not described in the eligibility criteria and exclusion criteria which could jeopardize the mentioned methodology and results.

•Lines 955-960: evaluating the quality of studies is not a prerequisite in scoping review. Thus; delete this paragraph.

Conclusion:

Major:

•Lines: 985-994: no need to describe the rational of conducting this review. Which was described in full in the introduction. Delete this paragraph.

•Lines: 995-1017: the conclusion is to restate the RQs and summarize the key findings results rather than redescribing the key findings or contrasting them with existing literature. Rewrite this section focus on summarize the key findings of this review, delete other information related to other literature.

General comments:

•The manuscript should be rewritten in a concise scientific language.

•The manuscript is disorganized; it should be rearranged in headings and subheadings that enhance its flow and readability and aligns with the Journal guidelines.

•Citation style is not aligned with the journal guidelines.

•the manuscript is wordy with a lot of redundancy; suggest to write in terse concise language.

•The manuscript should be thoroughly revised; some information missed in its section, however, described in wrong section.

6. PLOS authors have the option to publish the peer review history of their article (what does this mean?). If published, this will include your full peer review and any attached files.). If published, this will include your full peer review and any attached files.

.

Reviewer #1: **Yes:**Patrick ManserPatrick Manser

Reviewer #2: No

Reviewer #3: **Yes:**Dr. Suad J. GhabenDr. Suad J. Ghaben

While revising your submission, please upload your figure files to the Preflight Analysis and Conversion Engine (PACE) digital diagnostic tool, https://pacev2.apexcovantage.com/. PACE helps ensure that figures meet PLOS requirements. To use PACE, you must first register as a user. Registration is free. Then, login and navigate to the UPLOAD tab, where you will find detailed instructions on how to use the tool. If you encounter any issues or have any questions when using PACE, please email PLOS at . PACE helps ensure that figures meet PLOS requirements. To use PACE, you must first register as a user. Registration is free. Then, login and navigate to the UPLOAD tab, where you will find detailed instructions on how to use the tool. If you encounter any issues or have any questions when using PACE, please email PLOS at figures@plos.org. Please note that Supporting Information files do not need this step.. Please note that Supporting Information files do not need this step.

---

## [Author Response · Author response to Decision Letter 1]

16 May 2025

We have uploaded the "Response to Reviewers" document, which provides a tabulated set of responses to each reviewer point and is not easy to translate into this format. So we hope that is okay.

---

## [Decision Letter · Decision Letter 1]

5 Jun 2025

PONE-D-25-01664R1The Use of Digital Technology in Non-Pharmacological Cognitive and Psychosocial Interventions for People with Dementia and Mild Cognitive Impairment: A Scoping ReviewPLOS ONE

Dear Dr. Lowe,

Thank you for submitting your manuscript to PLOS ONE. After careful consideration, we feel that it has merit but does not fully meet PLOS ONE’s publication criteria as it currently stands. Therefore, we invite you to submit a revised version of the manuscript that addresses the points raised during the review process.

If applicable, we recommend that you deposit your laboratory protocols in protocols.io to enhance the reproducibility of your results. Protocols.io assigns your protocol its own identifier (DOI) so that it can be cited independently in the future. For instructions see: https://journals.plos.org/plosone/s/submission-guidelines#loc-laboratory-protocols. Additionally, PLOS ONE offers an option for publishing peer-reviewed Lab Protocol articles, which describe protocols hosted on protocols.io. Read more information on sharing protocols at . Additionally, PLOS ONE offers an option for publishing peer-reviewed Lab Protocol articles, which describe protocols hosted on protocols.io. Read more information on sharing protocols at https://plos.org/protocols?utm_medium=editorial-email&utm_source=authorletters&utm_campaign=protocols..

We look forward to receiving your revised manuscript.

Kind regards,

Mi-So Shim

Academic Editor

PLOS ONE

Additional Editor Comments :

Dear Authors,

Thank you for submitting the revised manuscript. Please find enclosed the reviewers’ comments on the revised version. We kindly ask you to review and address them accordingly.

Reviewers' comments:

Reviewer's Responses to Questions

**Comments to the Author**

1. If the authors have adequately addressed your comments raised in a previous round of review and you feel that this manuscript is now acceptable for publication, you may indicate that here to bypass the “Comments to the Author” section, enter your conflict of interest statement in the “Confidential to Editor” section, and submit your "Accept" recommendation.

Reviewer #1: (No Response)

Reviewer #2: All comments have been addressed

Reviewer #3: All comments have been addressed

2. Is the manuscript technically sound, and do the data support the conclusions?

Reviewer #1: Partly

Reviewer #2: Partly

Reviewer #3: Yes

3. Has the statistical analysis been performed appropriately and rigorously? 

Reviewer #1: N/A

Reviewer #2: Yes

Reviewer #3: N/A

4. Have the authors made all data underlying the findings in their manuscript fully available?

Reviewer #1: No

Reviewer #2: Yes

Reviewer #3: Yes

5. Is the manuscript presented in an intelligible fashion and written in standard English?

Reviewer #1: Yes

Reviewer #2: No

Reviewer #3: Yes

6. Review Comments to the Author

Reviewer #1: 1 Overview:

I appreciate the opportunity to review this revised manuscript and would like to thank the authors for their time and effort in addressing my initial evaluation. In my original review, I noted that the manuscript addresses a relevant research gap and presents an impressive volume of analyzed articles with findings that could meaningfully inform future research. Based on this, I had anticipated that the authors would be committed to implementing the recommendations provided to strengthen the completeness, methodological rigor, and overall impact of the review.

However, upon examining the response letter and the revised manuscript, it appears that while several comments were addressed thoughtfully, others appear to have been only partially incorporated or not reflected in the manuscript. Specifically, the authors provided a satisfactory response and made the necessary revisions for comments 4, 7, 9, 12 (as far as can be judged without visible track changes), and 15. Comments 2 and 10 were partially addressed, though there may still be opportunities to further enhance the reporting of results (comment 2) and to deepen the discussion in the Implications section (comment 10), particularly by integrating more nuanced reasoning and contextual links to the existing literature. For the remaining comments - nearly half of the total - the authors provided rebuttals; however, I was unable to identify corresponding revisions in the manuscript or its methods.

Most notably, my primary concern continues to center on the sensitivity of the search strategy and the screening methodology, which are foundational to the validity of a scoping review.

To elaborate, the review identified only 1982 records through the database search, which initially raised my concerns about the sensitivity of the search strategy. To assess whether this skepticism is substantiated by the current state of the evidence, I compared the review’s findings to those of recent reviews in related domains. According to Figure 3 of the manuscript, among others, 26 studies using gamified technologies, 26 studies with app-based interventions, 18 studies with desktop PC-based interventions, and 16 studies with VR-based interventions were included. In comparison, recent literature suggests that more relevant studies exist in the MCI or dementia population, given the observation that for each of these categories there are reviews available that identified more articles despite a narrower focus on specific populations and outcomes. For example:

Gamified technologies:

• Gao and Liu (2025) [1] identified 28 RCTs in the subcategory of “serious game-based interventions” targeting cognitive performance, although they restricted their analysis on MCI and serious games (which currently only account to about half of interventions in the exergaming research field).

App-based interventions:

• Silva et al (2024) [2] identified 34 studies in the subcategory of “app-based cognitive interventions” in their scoping review.

• Bateman et al. (2017) [3] identified 24 studies in the subcategory of “mobile app-based interventions” for individuals with MCI or dementia, although they restricted their analysis to community-dwelling populations and cognitive performance while this review is over 8 years old.

PC-based interventions:

• Chan et al. (2024) [4] identified 35 RCTs in the subcategory of “computerized cognitive training” in MCI and dementia, although they restricted their analyses to memory outcomes.

• Zuschnegg et al. (2023) [5] identified 24 RCTs in the subcategory of “computer-based interventions” for individuals with MCI or dementia, although they restricted their analysis to community-dwelling populations and cognitive performance

VR-based interventions:

• Chan et al. (2024) [6] identified 20 RCTs in the subcategory of “virtual-reality-based [serious and recreational] exergaming” , although they restricted their analysis to cognitive performance.

While some overlap in primary studies across these reviews is to be expected, this aligns with the double- or multi-categorization applied in the current scoping review. The observation that these reviews identified a similar or greater number of articles - despite narrower research questions (e.g., specific populations or outcomes) - supports my concern that this review may have missed relevant studies due to limitations in the search strategy.

Given the authors’ decision not to revise the search strategy as recommended - despite its central importance to the validity of the review’s findings - these unresolved concerns may limit the review’s comprehensiveness and reliability. While I appreciate the ambition of addressing such a broad and timely research question, reviews of this nature carry considerable weight in academic discourse and must therefore adhere to the highest standards of methodological rigor. For this reason, I believe the manuscript would benefit from further methodological refinement before it can be considered for publication. Should the authors be willing to refine the search strategy and re-screen the literature accordingly, I would be open to re-evaluating a substantially revised version.

2 References:

1. Gao Y, Liu N. Effects of digital technology-based serious games interventions for older adults with mild cognitive impairment: a meta-analysis of randomised controlled trials. Age and Ageing. 2025;54(4):afaf080. doi: https://doi.org/10.1093/ageing/afaf080.

2. Silva AF, Silva RM, et al. Cognitive Training with Older Adults Using Smartphone and Web-Based Applications: A Scoping Review. The Journal of Prevention of Alzheimer's Disease. 2024;11(3):693-700. doi: https://doi.org/https://doi.org/10.14283/jpad.2024.17.

3. Bateman DR, Srinivas B, et al. Categorizing Health Outcomes and Efficacy of mHealth Apps for Persons With Cognitive Impairment: A Systematic Review. Journal of Medical Internet Research. 2017;19(8):e301. doi: https://doi.org/10.2196/jmir.7814.

4. Chan ATC, Ip RTF, et al. Computerized cognitive training for memory functions in mild cognitive impairment or dementia: a systematic review and meta-analysis. npj Digital Medicine. 2024;7(1):1. doi: https://doi.org/10.1038/s41746-023-00987-5.

5. Zuschnegg J, Schoberer D, et al. Effectiveness of computer-based interventions for community-dwelling people with cognitive decline: a systematic review with meta-analyses. BMC Geriatrics. 2023;23(1):229. doi: https://doi.org/10.1186/s12877-023-03941-y.

6. Chan JYC, Liu J, et al. Exergaming and cognitive functions in people with mild cognitive impairment and dementia: a meta-analysis. npj Digital Medicine. 2024;7(1):154. doi: https://doi.org/10.1038/s41746-024-01142-4.

Reviewer #2: Based on the assessment of the manuscript "The Use of Digital Technology in Non-Pharmacological Cognitive and Psychosocial Interventions for People with Dementia and Mild Cognitive Impairment: A Scoping Review," I recommend Major Revision. The topic addresses a significant and timely research gap regarding digital interventions for dementia and MCI, and the scoping review methodology is generally appropriate for mapping this evolving field. The study identifies valuable patterns in technology types (e.g., prominence of gamified/app-based approaches, emergence of VR), geographical distribution (noting gaps in Africa/South America), and implementation settings (limited home-based studies). However, significant deficiencies currently preclude acceptance. Key concerns include: (1) Insufficient methodological transparency in screening/extraction processes (e.g., reviewer roles, conflict resolution, application of subjective "novelty" criteria, handling of overlapping technologies like GT/APP); (2) Problematic presentation and interpretation of results, particularly the oversimplified quantitative synthesis (Figs 7-9) using a "percentage change" metric that masks critical heterogeneity in outcomes, measures, and study designs, potentially leading to misleading conclusions about technology efficacy; (3) Inadequate discussion of the implications of extreme methodological heterogeneity across studies, limitations of the technology classification framework, and potential biases introduced by the novelty-focused inclusion criteria; and (4) Critical formatting issues in core tables (e.g., Table 1 data misalignment). Addressing these is essential for scientific rigor. The manuscript shows potential, but substantial revisions—particularly enhancing methodological reporting, replacing or rigorously contextualizing the quantitative synthesis, deepening the critical discussion of limitations/heterogeneity, and correcting presentation errors—are required before the work can be considered acceptable for publication. I encourage the authors to thoroughly address these points, as a revised version could make a valuable contribution to the literature.

Reviewer #3: (No Response)

7. PLOS authors have the option to publish the peer review history of their article (what does this mean?). If published, this will include your full peer review and any attached files.). If published, this will include your full peer review and any attached files.

.

Reviewer #1: **Yes:**Patrick ManserPatrick Manser

Reviewer #2: No

Reviewer #3: **Yes:**Suad J. GhabenSuad J. Ghaben

While revising your submission, please upload your figure files to the Preflight Analysis and Conversion Engine (PACE) digital diagnostic tool, https://pacev2.apexcovantage.com/. PACE helps ensure that figures meet PLOS requirements. To use PACE, you must first register as a user. Registration is free. Then, login and navigate to the UPLOAD tab, where you will find detailed instructions on how to use the tool. If you encounter any issues or have any questions when using PACE, please email PLOS at . PACE helps ensure that figures meet PLOS requirements. To use PACE, you must first register as a user. Registration is free. Then, login and navigate to the UPLOAD tab, where you will find detailed instructions on how to use the tool. If you encounter any issues or have any questions when using PACE, please email PLOS at figures@plos.org. Please note that Supporting Information files do not need this step.. Please note that Supporting Information files do not need this step.

---

## [Author Response · Author response to Decision Letter 2]

6 Mar 2026

We have uploaded our reviewer response letter. Please refer to this.

Above all we have updated the search strategy and also tried to eliminate text/figures that are suggestive of this being a systematic review rather than a scoping review.

---

## [Editor Report · Decision Letter 2]

16 Mar 2026

The Use of Digital Technology in Non-Pharmacological Cognitive and Psychosocial Interventions for People with Dementia and Mild Cognitive Impairment: A Scoping Review

PONE-D-25-01664R2

Dear Dr. Lowe,

We’re pleased to inform you that your manuscript has been judged scientifically suitable for publication and will be formally accepted for publication once it meets all outstanding technical requirements.

An invoice will be generated when your article is formally accepted. Please note, if your institution has a publishing partnership with PLOS and your article meets the relevant criteria, all or part of your publication costs will be covered. Please make sure your user information is up-to-date by logging into Editorial Manager at Editorial Manager® and clicking the ‘Update My Information' link at the top of the page. For questions related to billing, please contact  and clicking the ‘Update My Information' link at the top of the page. For questions related to billing, please contact billing support..

Kind regards,

Mi-So Shim, PhD

Academic Editor

PLOS One

---

## [Editor Report · Acceptance letter]

PONE-D-25-01664R2

PLOS One

Dear Dr. Lowe,

I'm pleased to inform you that your manuscript has been deemed suitable for publication in PLOS One. Congratulations! Your manuscript is now being handed over to our production team.

Kind regards,

on behalf of

Prof. Mi-So Shim

Academic Editor

PLOS One